# ROBUST SYSTEM IDENTIFICATION: NON-ASYMPTOTIC GUARANTEES AND CONNECTION TO REGULARIZATION

**Hyuk Park, Grani A. Hanasusanto, Yingying Li**
University of Illinois Urbana-Champaign
{hyukp2, gah, yl101}@illinois.edu

## ABSTRACT

We consider the problem of learning nonlinear dynamical systems from a single sample trajectory. While the least squares estimate (LSE) is commonly used for this task, it suffers from poor identification errors when the sample size is small or the model fails to capture the system's true dynamics. To overcome these limitations, we propose a robust LSE framework, which incorporates robust optimization techniques, and prove that it is equivalent to regularizing LSE using general Schatten $p$-norms. We provide non-asymptotic performance guarantees for linear systems, achieving an error rate of $\widetilde{\mathcal{O}}(1/\sqrt{T})$, and show that it avoids the curse of dimensionality, unlike state-of-the-art Wasserstein robust optimization models. Empirical results demonstrate substantial improvements in real-world system identification and online control tasks, outperforming existing methods.

## 1 INTRODUCTION

Many real-world problems require learning unknown dynamical systems from data. Examples can be from identifying the dynamics of mechanical systems like autonomous driving to predicting time-series data such as climate patterns and financial trends (Ng et al., 2006; Hong et al., 2008; Louka et al., 2008; Brunton et al., 2016; Alaskar, 2019). In the control community, the problem of estimating the parameters of a dynamical system is referred to as *system identification*. System identification is crucial since accurate estimation of underlying systems is integral to developing safe and reliable control systems.

In this work, we propose a robust system identification method for a certain class of nonlinear dynamical systems, assuming only a single trajectory of data is available. Specifically, the system is expressed as a linear combination of known nonlinear functions of the state and control inputs. Such system models have been widely applied since they accommodate a broad range of dynamic behaviors. One of the simplest system identification algorithms is the least squares method or the least squares estimate (LSE), which minimizes the squared prediction errors of the given samples. Due to the stochastic nature of the data, the performance of LSE cannot be deterministically guaranteed. Moreover, since the data comprises a single trajectory of states resulting from the evolution of the dynamical system, the samples are non-i.i.d. Recent works (Simchowitz et al., 2018; Jedra & Proutiere, 2020) provide non-asymptotic analyses of LSE, specifically addressing system identification errors with respect to a finite number of non-i.i.d. samples. They show that the error decays as fast as the optimal rate $\mathcal{O}(1/\sqrt{T})$ where $T$ denotes the number of samples.

Although these theoretical results are promising, the empirical performance of LSE may suffer, particularly when only a few samples are available or the model is misspecified. This limitation is critical in applications where data collection is inherently restricted, or the true dynamics are highly complex. To address these issues, we propose a robust approach that combines robust optimization with LSE by formulating a min-max optimization problem, referred to as the robust LSE problem. We show that the robust LSE problem can be cast as a convex semidefinite program (SDP), making it tractable to solve. Additionally, we provide a non-asymptotic analysis for our approach and demonstrate that robust LSE achieves a near-optimal error rate of $\widetilde{\mathcal{O}}(1/\sqrt{T})$. Interestingly, we show that our robust LSE problem is equivalent to the LSE problem with an additional regularization term

based on the general Schatten $p$-norm. While a few special cases of Schatten $p$-norms have been used to regularize LSE problems (Abbasi-Yadkori & Szepesvári, 2011; Sun et al., 2022), these methods do not guarantee asymptotic convergence to the true system parameters under a single trajectory. To our knowledge, our work is the first to provide a non-asymptotic analysis for LSE with general Schatten $p$-norm regularization under a single trajectory.

An alternative data-driven robust approach to ours is the state-of-the-art Wasserstein robust optimization (Mohajerin Esfahani & Kuhn, 2018), which has gained considerable attention in the machine learning community for its promising performance (Shafieezadeh Abadeh et al., 2018; Liu et al., 2022; Nietert et al., 2024; Bai et al., 2024). However, this approach suffers from the curse of dimensionality: for systems with a high-dimensional state space, its error rate decays slowly.

The contributions of this paper are summarized as follows:

i. We introduce a novel system identification algorithm that combines robust optimization with the LSE framework. Additionally, we establish the equivalence between our robust LSE problem and the LSE problem regularized by the Schatten $p$-norm. This is significant for the regularization framework because, as noted in Abu-Mostafa et al. (2012), "*most of the regularization methods used successfully in practice are heuristic methods.*" The equivalence, however, enables regularization to borrow a probabilistic interpretation from robust optimization, suggesting that the regularization term should be data-dependent to ensure good out-of-sample (i.e., test) performance.

ii. We provide a theoretical performance guarantee for our robust method, achieving an error rate of $\widetilde{\mathcal{O}}(1/\sqrt{T})$. This result is notable not only as the first performance guarantee for regularized LSE under the single trajectory setting but also because it shows that our robust LSE circumvents the curse of dimensionality, unlike the emerging data-driven Wasserstein robust optimization models—hence, offering new insights into the robust regression literature.

iii. We conduct numerical experiments that demonstrate substantial performance improvements in real-world system identification tasks, such as short-term wind speed prediction. We also conduct extensive experiments with synthetic dynamical systems to validate our approach. Additionally, we showcase its effectiveness in online control tasks by integrating our robust LSE with existing online linear quadratic control algorithms, demonstrating consistently better performance compared to existing methods.

FUTHER LITERATURE REVIEW

There has been a recent emergence of interest in deriving non-asymptotic systems identification errors. Most works focus on analyzing performance of the *standard* LSE (Simchowitz et al., 2018; Faradonbeh et al., 2018; Sarkar & Rakhlin, 2019; Mania et al., 2019; Foster et al., 2020; Dean et al., 2020; Jedra & Proutiere, 2020; Sattar et al., 2021; Kowshik et al., 2021; Sattar & Oymak, 2022; Mania et al., 2022; Li et al., 2023). One advantage of analyzing the standard LSE is that the system identification error term, which is the main interest of the analysis, can be analytically obtained using the solution to the LSE problem. This term can then be broken down in various ways, enabling different approaches to address the resulting components.

While theoretical guarantees for LSE appear promising, the empirical performance degrades in real-world applications where available data is scarce, resulting in subpar estimates (Sun et al., 2022). We employ robust optimization techniques (Ben-Tal et al., 2009) to enhance the resilience of LSE. The key idea of robust optimization is to find solutions that perform optimally against the worst-case realizations of uncertain data. Dean et al. (2020) assume i.i.d. samples and utilize the standard LSE for system identification. They construct an uncertainty set of system parameters around the resulting estimate. They then solve a min-max problem, referred to as the robust LQR problem, to determine the best control input against the worst-case system parameter in the uncertainty set. In contrast, our approach directly formulates a min-max problem for system identification, seeking the best system parameter against the worst-case realizations of the data. To the best of our knowledge, no prior work has proposed a robust LSE problem formulation presented in this paper.

As stated in our contributions, the non-asymptotic analysis proposed in this paper is not limited to our robust LSE. It can be extended to the regularized LSE problem, where the regularization term is defined as the Schatten $p$-norm of a quadratic function of system parameters multiplied by

a user-defined (scalar) tuning parameter—henceforth referred to as the regularization parameter. Special cases of the Schatten $p$-norm regularization are proposed in the literature (Abbasi-Yadkori & Szepesvári, 2011; Sun et al., 2022). In Abbasi-Yadkori & Szepesvári (2011), they introduce the squared Frobenius norm of system parameters with the regularization parameter set to some small *fixed* value, hence the convergence of their estimate to the true system parameter is not guaranteed. In Sun et al. (2022), they consider systems with limited state observations, i.e., states cannot be directly observed. They introduce the nuclear norm of the Hankel matrix to their LSE problem and derive the non-asymptotic impulse response estimation errors only for the MISO (multi-input single-output) system under the assumption that i.i.d. samples are available. These papers do not provide guidance on how to adjust the regularization parameter as a function of the number of available samples.

NOTATION

Bold lower-case letter $\boldsymbol{x}$ and upper-case letter $\boldsymbol{X}$ represent a vector and a matrix, respectively, while regular font $x$ indicates a scalar. For any square matrix $\boldsymbol{X} \in \mathbb{R}^{n \times n}$, the trace operation is denoted as $\mathrm{tr}(\boldsymbol{X})$. The spectral radius denoted $\rho(\boldsymbol{X})$ is the largest absolute value of the eigenvalues of $\boldsymbol{X}$. An $n \times n$ dimensional identity matrix is denoted as $\mathbb{I}_n$. For any $n \times m$ matrix $\boldsymbol{X} \in \mathbb{R}^{n \times m}$, $\|\boldsymbol{X}\|_p$ represents the Schatten $p$-norm of the matrix, which is defined as $\|\boldsymbol{X}\|_p = (\mathrm{tr}(\boldsymbol{X}^\top \boldsymbol{X})^{p/2})^{1/p}$. For several special cases of the Schatten $p$-norm, we may interchangeably use the following notations: nuclear norm $\|\cdot\|_* = \|\cdot\|_1$, Frobenius norm $\|\cdot\|_F = \|\cdot\|_2$, and operator norm $\|\cdot\| = \|\cdot\|_\infty$. In the entire paper, we will *not* use matrix norms induced by vector norms to prevent any confusion. We may use $\|\boldsymbol{x}\|_2$ for the Euclidean norm (i.e., $\ell^2$ norm). In this case, the notation should still be clear since the norm is taken on a vector (i.e., bold lower-case). To denote an $n \times n$ positive (semi)definite matrix $\boldsymbol{X}$, we may interchangeably use $\boldsymbol{X} \in \mathbb{S}^n_{++}$ ($\boldsymbol{X} \in \mathbb{S}^n_+$) and $\boldsymbol{X} \succ 0$ ($\boldsymbol{X} \succeq 0$). We use the standard $\mathcal{O}(\cdot)$ notation to describe the complexity of a function. In addition, $\widetilde{\mathcal{O}}(\cdot)$ is used to suppress multiplicative terms with logarithmic dependence.

## 2 PROBLEM STATEMENT

Consider the following class of nonlinear dynamical system models

$$\boldsymbol{x}_{t+1} = \boldsymbol{\theta}^\star \phi(\boldsymbol{x}_t, \boldsymbol{u}_t) + \boldsymbol{w}_t, \quad t = 0, \dots, T-1. \tag{1}$$

Let $\boldsymbol{x}_t \in \mathbb{R}^n$, $\boldsymbol{u}_t \in \mathbb{R}^m$, and $\boldsymbol{w}_t \in \mathbb{R}^n$ represent the state, control input, and noise at time $t$, respectively. The feature map $\phi : \mathbb{R}^n \times \mathbb{R}^m \to \mathbb{R}^p$ is an arbitrary, known nonlinear function. To simplify notation, we use the feature map interchangeably as $\phi(\boldsymbol{x}_t, \boldsymbol{u}_t)$ and $\phi(\boldsymbol{z}_t)$, where $\boldsymbol{z}_t = [\boldsymbol{x}_t^\top \ \boldsymbol{u}_t^\top]^\top \in \mathbb{R}^{n+m}$ is the augmented vector of the state and control input.

Our goal is to recover the unknown parameters $\boldsymbol{\theta}^\star \in \mathbb{R}^{n \times p}$ from a single trajectory of data. The model (1) is versatile enough to capture a broad range of real-world applications, from mechanical systems like autonomous helicopters and bipedal robots to time-series models commonly used in financial markets, weather prediction, and epidemiology for modeling disease spread (Ljung, 1998; Ng et al., 2006; Hong et al., 2008; Louka et al., 2008; Brunton et al., 2016; Alaskar, 2019).

### 2.1 LEAST SQUARES ESTIMATE

The least squares estimate (LSE) is widely used for system identification. Given a single trajectory $(\{\boldsymbol{z}_t\}_{t=0}^{T-1}, \boldsymbol{x}_T)$, the LSE denoted as $\overline{\boldsymbol{\theta}}_T$ minimizes the sum of the squares of the residuals:

$$\overline{\boldsymbol{\theta}}_T \in \arg\min_{\boldsymbol{\theta}} \frac{1}{T} \sum_{t=0}^{T-1} \|\boldsymbol{x}_{t+1} - \boldsymbol{\theta}\phi(\boldsymbol{z}_t)\|_2^2. \tag{2}$$

Let us refer to the minimization in (2) as the LSE problem. Note that the objective function in (2) is quadratic in $\boldsymbol{\theta}$. Therefore, we can rewrite the LSE problem as

$$\min_{\boldsymbol{\theta}} \frac{1}{T} \sum_{t=0}^{T-1} \begin{bmatrix} \boldsymbol{x}_{t+1} \\ \phi(\boldsymbol{z}_t) \end{bmatrix}^\top \begin{bmatrix} \mathbb{I}_n & -\boldsymbol{\theta} \\ -\boldsymbol{\theta}^\top & \boldsymbol{\theta}^\top\boldsymbol{\theta} \end{bmatrix} \begin{bmatrix} \boldsymbol{x}_{t+1} \\ \phi(\boldsymbol{z}_t) \end{bmatrix} = \min_{\boldsymbol{\theta}} \ \mathrm{tr}\left(\boldsymbol{G}(\boldsymbol{\theta})\widehat{\boldsymbol{\Omega}}_T\right), \tag{3}$$

$$\text{where } \boldsymbol{G}(\boldsymbol{\theta}) = \begin{bmatrix} \mathbb{I}_n & -\boldsymbol{\theta} \\ -\boldsymbol{\theta}^\top & \boldsymbol{\theta}^\top\boldsymbol{\theta} \end{bmatrix} \text{ and } \widehat{\boldsymbol{\Omega}}_T = \frac{1}{T}\sum_{t=0}^{T-1} \begin{bmatrix} \boldsymbol{x}_{t+1} \\ \phi(\boldsymbol{z}_t) \end{bmatrix} \begin{bmatrix} \boldsymbol{x}_{t+1} \\ \phi(\boldsymbol{z}_t) \end{bmatrix}^\top. \tag{4}$$

We can express the *true* LSE problem by substituting $\widehat{\boldsymbol{\Omega}}_T$ in (3) with its expectation, namely, $\boldsymbol{\Omega}_T^\star = \mathbb{E}[\widehat{\boldsymbol{\Omega}}_T]$. Then, one can obtain the system parameter by solving the true LSE problem:

$$\boldsymbol{\theta}^\star \in \arg\min_{\boldsymbol{\theta}} \ \operatorname{tr}\left(\boldsymbol{G}(\boldsymbol{\theta})\boldsymbol{\Omega}_T^\star\right). \tag{5}$$

From (5), it is clear that obtaining the true system parameter requires knowledge of $\boldsymbol{\Omega}_T^\star$, while the empirical estimate $\widehat{\boldsymbol{\Omega}}_T$ inherently contains estimation errors that depend on the available data. In other words, a poor estimate $\widehat{\boldsymbol{\Omega}}_T$ may lead to inferior performance, which is commonly the case when the sample size $T$ is small or, in our context, a short trajectory of data.

## 3 ROBUST LEAST SQUARES ESTIMATE

As mentioned earlier, the estimate $\widehat{\boldsymbol{\Omega}}_T$ based on $T$ non-i.i.d. samples may fail to accurately capture $\boldsymbol{\Omega}_T^\star$ when $T$ is small. Even with a sufficiently large $T$, the standard LSE $\bar{\boldsymbol{\theta}}_T$ in (2) may still perform poorly in practical applications where the model (1) does not adequately reflect the true behavior of the dynamical system. To address this issue, we formulate a robust version of the LSE problem to obtain a robust estimate, denoted as $\widehat{\boldsymbol{\theta}}_T$:

$$\widehat{\boldsymbol{\theta}}_T \in \arg\min_{\boldsymbol{\theta}} \ \max_{\boldsymbol{\Omega}\in\mathcal{U}_T^{p,\epsilon}} \operatorname{tr}\left(\boldsymbol{G}(\boldsymbol{\theta})\boldsymbol{\Omega}\right) \text{ where } \mathcal{U}_T^{p,\epsilon} = \left\{\boldsymbol{\Omega}\in\mathbb{S}_+^{2n+m} : \|\boldsymbol{\Omega}-\widehat{\boldsymbol{\Omega}}_T\|_p \le \epsilon\right\}. \tag{6}$$

The proposed approach (6) first constructs the uncertainty set $\mathcal{U}_T^{p,\epsilon}$ which contains all positive semidefinite matrices $\boldsymbol{\Omega}$ that are within a distance of $\epsilon \ge 0$ from the estimate $\widehat{\boldsymbol{\Omega}}_T$ in the Schatten $p$-norm. Then, it seeks a minimizer $\widehat{\boldsymbol{\theta}}_T$ that performs best under the worst-case matrix $\boldsymbol{\Omega}$ in $\mathcal{U}_T^{p,\epsilon}$. However, the min-max problem in (6) is difficult to solve directly since the objective function involves a maximization problem. In the following, we introduce an equivalent semidefinite program (SDP) for the robust LSE problem. To our knowledge, the proposed formulation has not been derived in the literature of robust regression problems.

**Theorem 1.** *For any given uncertainty set parameters $p \ge 1$ (as in the Schatten $p$-norm) and $\epsilon \ge 0$, the robust LSE problem in (6) can be equivalently reformulated as the SDP*

$$\begin{aligned} \min \quad & \operatorname{tr}(\boldsymbol{\Gamma}\widehat{\boldsymbol{\Omega}}_T) + \epsilon\|\boldsymbol{\Gamma}\|_q \\ \text{s.t.} \quad & \boldsymbol{\theta}\in\mathbb{R}^{n\times(n+m)}, \quad \boldsymbol{\Gamma}\in\mathbb{S}_+^{2n+m}, \quad \boldsymbol{H}\in\mathbb{S}_+^{n+m}, \\ & \boldsymbol{\Gamma}\succeq\begin{bmatrix} \mathbb{I}_n & -\boldsymbol{\theta} \\ -\boldsymbol{\theta}^\top & \boldsymbol{H} \end{bmatrix}, \\ & \begin{bmatrix} \mathbb{I}_n & \boldsymbol{\theta} \\ \boldsymbol{\theta}^\top & \boldsymbol{H} \end{bmatrix} \succeq \boldsymbol{0}, \end{aligned} \tag{7}$$

*where $\|\cdot\|_q$ is the dual Schatten norm of $\|\cdot\|_p$, that is, $q$ such that $\frac{1}{p}+\frac{1}{q}=1$.*

The proof of this theorem is in the supplementary material A.

Note that the Schatten $p$-norm defined in (6) corresponds to the Schatten $q$-norm in the objective function in (7). For any $q \ge 1$, the reformulation (7) is a convex SDP. In particular, for several choices of $q$ such as $q = 1, 2, \infty$, it is readily solvable by off-the-shelf commercial solvers. Moreover, the computational complexity of our approach is invariant to the number of samples $T$, i.e., the size of the SDP (7) remains the same regardless of $T$ since the model only requires the matrix $\widehat{\Omega}_T$.

Interestingly, the robust LSE problem admits an equivalence to the regularized LSE problem as shown in the following corollary.

**Corollary 1.** *For any given uncertainty set parameters $p \ge 1$ and $\epsilon \ge 0$, the robust LSE problem (6) is equivalent to the LSE problem with the Schatten $q$-norm regularization term as follows:*

$$\min_{\boldsymbol{\theta}} \operatorname{tr}\left(\boldsymbol{G}(\boldsymbol{\theta})\widehat{\boldsymbol{\Omega}}_T\right) + \epsilon\|\boldsymbol{G}(\boldsymbol{\theta})\|_q. \tag{8}$$

The proof of this corollary is in the supplementary material B.

A few remarks are in order. If the nuclear norm (i.e., $q = 1$) is used in (8), then we have $\epsilon\|\boldsymbol{G}(\boldsymbol{\theta})\|_* = \epsilon\|\boldsymbol{\theta}\|_F^2 + \epsilon n$. Thus, the regularization term simplifies to a squared Frobenius norm regularization on $\boldsymbol{\theta}$, and the resulting problem constitutes a tractable quadratic program. Corollary 1 further draws an interesting connection between the robust LSE and the regularized LSE in Abbasi-Yadkori & Szepesvári (2011). In that work, the regularization parameter $\epsilon$ is set to a small *fixed* value. However, it lacks a clear explanation of how the regularization impacts the performance of the LSE since it is introduced merely to ensure the invertibility of the matrix $\sum_{t=0}^{T-1} \boldsymbol{x}_{t+1}\boldsymbol{x}_{t+1}^\top$ (known as *the Gram matrix* as discussed in the following section). In this case, a convergence rate on the system identification error cannot be established. Our result, therefore, not only provides a justification for the use of squared Frobenius norm regularization but also guidance on how to control the parameter as the sample size $T$ increases. The recent work Sun et al. (2022) uses a Hankel nuclear norm regularization to identify low-order linear systems using i.i.d. trajectories. They recognize that the regularization term yields better performance than the unregularized LSE when the number of samples is small. However, to the best of our knowledge, there is no prior non-asymptotic analysis for the LSE with general Schatten norm regularization under the single trajectory assumption. In Section 4, we provide non-asymptotic analyses for the robust LSE problem, which ultimately results in system identification errors.

## 4 PERFORMANCE GUARANTEES

Among the simplest examples of (1) is the linear system, where the feature map is defined as $\phi(\boldsymbol{x}_t, \boldsymbol{u}_t) = [\boldsymbol{x}_t^\top \ \boldsymbol{u}_t^\top]^\top \in \mathbb{R}^{n+m}$. In this case, the system evolves according to $\boldsymbol{x}_{t+1} = \boldsymbol{A}^\star\boldsymbol{x}_t + \boldsymbol{B}^\star\boldsymbol{u}_t + \boldsymbol{w}_t$, where the unknown parameters are $\boldsymbol{\theta}^\star = [\boldsymbol{A}^\star \ \boldsymbol{B}^\star] \in \mathbb{R}^{n \times (n+m)}$. In the literature, much of the statistical analysis concerning LSE performance focuses on understanding the sum of outer products, $\sum_{t=0}^{T-1} \boldsymbol{x}_{t+1}\boldsymbol{x}_{t+1}^\top$, known as the Gram matrix. Suppose that the noise is Gaussian, e.g., $\boldsymbol{w}_t \sim \mathcal{N}(\boldsymbol{0}, \Sigma_w)$, and the sequence of control inputs is generated by a Gaussian distribution, e.g., $\boldsymbol{u}_t \sim \mathcal{N}(\boldsymbol{0}, \sigma_u^2\mathbb{I})$ for $t = 0, \ldots, T-1$. Then, the expected Gram matrix $\mathbb{E}[\sum_{t=0}^{T-1} \boldsymbol{x}_{t+1}\boldsymbol{x}_{t+1}^\top]$, which corresponds to the first diagonal block of $\boldsymbol{\Omega}_T^\star$ in (5), can be nicely represented as a matrix-valued function of the unknown system parameter $\boldsymbol{\theta}^\star$:

$$\mathbb{E}\left[\sum_{t=0}^{T-1} \boldsymbol{x}_{t+1}\boldsymbol{x}_{t+1}^\top\right] = \sum_{t=0}^{T-1} \boldsymbol{\Gamma}_t(\boldsymbol{\theta}^\star) = \sum_{t=0}^{T-1}\sum_{s=0}^{t}(\boldsymbol{A}^{\star^\top})^s(\sigma_u^2\boldsymbol{B}^\star\boldsymbol{B}^{\star\top} + \Sigma_w)(\boldsymbol{A}^\star)^s. \tag{9}$$

Of course, the *expected* Gram matrix (9) is not accessible to us since it requires $\boldsymbol{\theta}^\star$.

In this section, we discuss the non-asymptotic guarantees of our robust approach for *linear* systems under the single trajectory assumption. Specifically, our goal is to show that the system identification errors of our robust method matches the near-optimal rate $\widetilde{\mathcal{O}}(1/\sqrt{T})$. This suggests that introducing robustness incurs only negligible costs in terms of $T$, while providing significant empirical improvements over the *unregularized* LSE in (2) (henceforth referred to as the *standard* LSE), as discussed in the following section.

As noted in Tsiamis et al. (2023), despite its apparent simplicity, the linear system remains challenging to analyze. The majority of research in statistical learning for system identification has focused on linear systems, as this setting allows for more tractable theoretical analysis. While a few papers have analyzed certain classes of nonlinear systems, these often sidestep the core challenges by focusing on systems that exhibit near-linear behavior (Foster et al., 2020; Sattar et al., 2021; Kowshik et al., 2021; Sattar & Oymak, 2022; Mania et al., 2022).

For the standard LSE, many works have established optimal rates of convergence. A key advantage of analyzing the standard LSE is that the system identification error can be directly derived from the analytical solution to the LSE problem in (2). Specifically, the error is given by $\bar{\boldsymbol{\theta}}_T - \boldsymbol{\theta}^\star = (\sum_{t=0}^{T-1} \boldsymbol{w}_t\boldsymbol{z}_t^\top)(\sum_{t=0}^{T-1} \boldsymbol{z}_t\boldsymbol{z}_t^\top)^{-1}$. This expression allows the error term to be decomposed in various ways to enable different types of analysis (Simchowitz et al., 2018; Sarkar & Rakhlin, 2019; Jedra & Proutiere, 2020). However, these decomposition techniques do not apply to our robust LSE, as the error term for the robust estimator, i.e., $\widehat{\boldsymbol{\theta}}_T - \boldsymbol{\theta}^\star$, no longer has a convenient analytical form. Therefore, a different approach is required for our analysis.

## 4.1 ASSUMPTIONS

We formally state our assumptions for the analysis in this section.

**A1.** We consider a strictly stable system, i.e., $\rho(\boldsymbol{A}^\star) < 1$.

**A2.** The data, i.e., observed states of the system (1), is collected in a single trajectory of length $T + 1$ denoted as $\{\boldsymbol{x}_t\}_{t=0}^T \in \mathbb{R}^{n(T+1)}$ with the initial state $\boldsymbol{x}_0 = \boldsymbol{0}$.

**A3.** Let $\{\mathcal{F}_t\}_{t \geq 0}$ be a filtration and $\{\boldsymbol{x}_t\}_{t \geq 0}$ be a stochastic process such that $\boldsymbol{x}_t$ is $\mathcal{F}_{t-1}$ measurable.

**A4.** The noise $\boldsymbol{w}_t \in \mathbb{R}^n$ is a martingale difference sequence with respect to $\mathcal{F}_t$ with $\mathbb{E}[\boldsymbol{w}_t | \mathcal{F}_{t-1}] = \boldsymbol{0}$ and $\mathbb{E}[\boldsymbol{w}_t \boldsymbol{w}_t^\top | \mathcal{F}_{t-1}] = \boldsymbol{\Sigma}_w \succ \boldsymbol{0}$.

**A5.** Furthermore, we assume that $\boldsymbol{w}_t$ is a $\sigma_w^2$-conditionally sub-Gaussian random vector with respect to $\mathcal{F}_t$, i.e., for any unit vector $\boldsymbol{v} \in \mathbb{R}^n$, there exists some constant $\sigma_w^2 > 0$ such that the inner-product $\boldsymbol{v}^\top \boldsymbol{w}_t$ follows the inequality $\mathbb{E}[\exp(\boldsymbol{v}^\top \boldsymbol{w}_t) | \mathcal{F}_t] \leq \exp(\|\boldsymbol{v}\|^2 \sigma_w^2 / 2)$.

**A6.** The control input $\boldsymbol{u}_t \in \mathbb{R}^m$ is an i.i.d. $\sigma_u^2$-sub-Gaussian random vector with $\mathbb{E}[\boldsymbol{u}_t] = \boldsymbol{0}$ and $\mathbb{E}[\boldsymbol{u}_t \boldsymbol{u}_t^\top] = \sigma_u^2 \mathbb{I}_m$. In other words, we inject sub-Gaussian exploration noise into the system to identify the system parameter $\boldsymbol{\theta}^\star$.

These are standard assumptions in the literature. In particular, **A3.-A5.** enable us to utilize tools from the self-normalized process (Abbasi-Yadkori et al., 2011). The main challenge in our analysis arises from the single trajectory assumption in **A2.**, as this trajectory consists of non-i.i.d. samples. Due to this difficulty, some previous works rely on a more stringent assumption that $T$ multiple i.i.d. trajectories are available, taking only the last state from each trajectory to ensure that those $T$ samples are i.i.d. Our main theoretical contribution lies in deriving non-asymptotic guarantees of the proposed robust LSE method (6), using non-i.i.d. samples.

We first provide the non-asymptotic coverage guarantee of our uncertainty set in (6), which is eventually used as the main ingredient for our system identification error analysis.

**Proposition 1.** *(Non-asymptotic coverage guarantee). For any significance level $\delta \in (0, 1]$, we have*

$$\mathbb{P}\left[\boldsymbol{\Omega}_T^\star \in \mathcal{U}_T^p(\delta)\right] \geq 1 - \delta. \tag{10}$$

*Here, $\mathcal{U}_T^p(\delta)$ is the uncertainty set for the robust LSE problem in (6) defined as follows:*

$$\mathcal{U}_T^p(\delta) = \left\{ \boldsymbol{\Omega} \in \mathbb{S}_+^{2n+m} : \|\boldsymbol{\Omega} - \widehat{\boldsymbol{\Omega}}_T\|_p \leq \epsilon(\delta) \right\} \text{ and } \epsilon(\delta) = \widetilde{\mathcal{O}}(1/\sqrt{T}). \tag{11}$$

The proof of this proposition is in the supplementary material C.

Before discussing the main results, we make several comments about Proposition 1. Although (37) provides a more explicit form for the upper bound $\epsilon(\delta)$ in (11) by identifying the universal constants, it would be an overly conservative estimate and thus lack practical usage by itself. Instead of relying on such a conservative a priori bound, it is common practice to calibrate the regularization parameter, using the cross-validation procedure (Arlot & Celisse, 2010; Mohajerin Esfahani & Kuhn, 2018; Shafieezadeh Abadeh et al., 2018; Bach, 2024). In the following section, we will use cross-validation to select the initial regularization parameter. In this context, Proposition 1 becomes practically useful, as the rate $\widetilde{\mathcal{O}}(1/\sqrt{T})$ offers guidance on how to scale the regularization parameter of the robust LSE as the sample size increases. We believe that these results are insightful from a broad range of perspectives, including machine learning, system identification, and robust optimization.

In the system identification literature, regularization is often applied merely to ensure strong convexity, with the regularization parameter typically set heuristically to a small value (Abbasi-Yadkori & Szepesvári, 2011; Sun et al., 2022). Our results, however, suggest that the regularization parameter should be data-dependent to achieve good out-of-sample performance guarantees. To the best of our knowledge, there are no existing works that provide a theoretical analysis of the regularized LSE under a single trajectory.

In the literature on robust optimization, Mohajerin Esfahani & Kuhn (2018) introduces the state-of-the-art Wasserstein robust optimization model under the i.i.d. data setting. While their model

could, in principle, be an alternative to ours, unfortunately, their analysis reveals that the Wasserstein model suffers from the curse of dimensionality, i.e., their error rate $\mathcal{O}(1/T^{\frac{2}{n}})$ becomes slower as the dimension of the state space, $n$, increases. Although a more recent work Gao (2023) addresses this challenge by deriving *dimension-free* non-asymptotic guarantees under the Lipschitz continuity assumption with respect to $\boldsymbol{\theta}$, it may not easily generalize and, more importantly, it is not applicable to our problem. While the Wasserstein model has garnered significant attention in the machine learning community for its promising performance across various applications (Shafieezadeh Abadeh et al., 2018; Liu et al., 2022; Bai et al., 2024; Nietert et al., 2024), there is still an open question about whether the error rate can be improved. In this regard, our results provide new insights into the open question by avoiding the curse of dimensionality, even under the more stringent assumption of non-i.i.d. data from a single trajectory.

Building upon the insights from Proposition 1, we now turn to the analysis of the non-asymptotic system identification errors. Specifically, the following theorem applies to both the robust LSE and the regularized LSE, due to their equivalence.

**Theorem 2.** *(System identification errors). Suppose that $\epsilon(\delta)$ is the upper bound in (11). Then, for any significance level $\delta \in (0, 1]$, as long as*

$$T \geq T(\delta) = \left( \frac{400}{3} \right) \left( \log \left( \frac{1}{\delta} \right) + 2(n + m) \log \left( \frac{200}{3} \right) + \log\det \left( \widetilde{\boldsymbol{\Gamma}}_1 \overline{\boldsymbol{\Gamma}}_1^{-1} \right) \right)$$

*where*

$$\widetilde{\boldsymbol{\Gamma}}_1 = \left[ \begin{array}{cc} \boldsymbol{\Gamma}_1(\boldsymbol{\theta}^\star) & \mathbf{0} \\ \mathbf{0} & \sigma_u^2 \mathbb{I}_m \end{array} \right] \quad and \quad \overline{\boldsymbol{\Gamma}}_1 = \frac{n + m}{\delta} \mathbb{E} \left[ \boldsymbol{z}_1 \boldsymbol{z}_1^\top \right],$$

*we have the following system identification errors*

$$\mathbb{P} \left[ \|\boldsymbol{\theta}^\star - \widehat{\boldsymbol{\theta}}_T\| \leq \frac{\epsilon(\delta) \sqrt{\min\{n, m\}}}{\widehat{\alpha}} (2 + 2\|\boldsymbol{\theta}^\star\| + \|\nabla_{\boldsymbol{\theta}}\|\boldsymbol{G}(\boldsymbol{\theta}^\star)\|_q\|) \right] \geq 1 - \delta, \qquad (12)$$

*where $\widehat{\alpha} = \frac{1}{24} \left( \frac{3}{20} \right)^2 \min \left\{ \sigma_w^2, \sigma_u^2 \right\}$.*

The proof of this theorem is in the supplementary material D.

## 5 NUMERICAL EXPERIMENTS

In this section, we present numerical experiments to evaluate the performance of our proposed method. Both the proposed approach and benchmark models are implemented in Python 3.7. Specifically, a neural network model is implemented using TensorFlow (Abadi et al., 2015), while the optimization problem (7) is modeled with the CVXPY (Diamond & Boyd, 2016) interface and solved using the commercial solver MOSEK (ApS, 2024). All experiments were conducted on a laptop equipped with a 6-core, 2.3 GHz Intel Core i7 CPU and 16 GB of RAM. The SDP formulations for the examples in this section can be solved in under 0.1 seconds. In the supplementary material E, we provide the mean computational times for several example systems.

We compare our robust LSE with the standard LSE for the wind speed prediction problem (Tascikaraoglu & Uzunoglu, 2014) and learning synthetic dynamical systems. Additionally, we consider an online control task where we combine our robust LSE with the existing online linear quadratic (LQ) control algorithms. We then compare the regret of different algorithms to demonstrate how improved performance in system identification can be translated into more reliable control systems.

As commented earlier, while the theoretical error rate $\widetilde{\mathcal{O}}(1/\sqrt{T})$ derived in Proposition 1 is still useful, choosing the regularization parameter directly from the theoretical upper bound $\epsilon(\delta)$ leads to a too conservative estimate $\widehat{\boldsymbol{\theta}}_T$. In fact, a similar argument is made in Dean et al. (2020). Instead of adopting the theoretical guarantee, the authors use the standard bootstrap method to obtain an empirical upper bound $\overline{\epsilon}(\delta)$ on the system identification errors of the standard LSE, i.e., $\|\boldsymbol{\theta}^\star - \overline{\boldsymbol{\theta}}_T\| \leq \overline{\epsilon}(\delta)$. For the robust LSE, we use a 3-fold cross-validation procedure to determine an initial value of the regularization parameter, as follows. We split the samples into three equal-sized subsets where two of the three subsets are put together to learn the robust estimate. The resulting estimate is then tested on the remaining set for all $\epsilon = (a \cdot 10^b)/\sqrt{T}$ where $a \in \{1, 3, 5, 7, 9\}$ and $b \in \{-3, \ldots, 3\}$. This process is repeated three times for different partitions of the samples to choose the $\epsilon$ that performs best overall.

## 5.1 WIND SPEED PREDICTION

We address the wind speed prediction problem as an example of learning an underlying nonlinear time-series model. Accurate wind speed prediction is essential for the safe integration of wind energy into power grids. However, the nonlinear nature of wind speed makes this task particularly challenging and an active research area, with various approaches being proposed, including physics-based models and neural network-based designs (Louka et al., 2008; Liu et al., 2018; Chen et al., 2021; Cai et al., 2021; Theuer et al., 2021; Hazarika et al., 2022). A recent work Chen et al. (2024) decomposes the raw wind speed data into simpler nonlinear components known as intrinsic mode functions (IMFs) using the Hilbert–Huang transform (HHT) based on complementary ensemble empirical mode decomposition. Neural network models are employed to learn the IMFs, and the standard LSE is used to determine the optimal weights for these learned nonlinear functions.

Note that the nonlinear function $\phi(\cdot)$ is not given explicitly in this experiment but is learned via the neural network model, making the problem more challenging than our problem setup. We chose the wind speed prediction problem because a successful implementation would demonstrate that complex nonlinear systems can be effectively learned by combining machine learning and optimization methods without extensive domain knowledge. Using the wind speed data from fedesoriano (2022), we implemented the optimized HHT-NAR method from Chen et al. (2024) (here simply referred to as *LSE*), along with our robust version, and evaluated the prediction accuracy for the next 50 daily wind speeds. Figure 1 shows prediction results for both methods using a single trajectory of 30 sample points (i.e., $T = 30$).

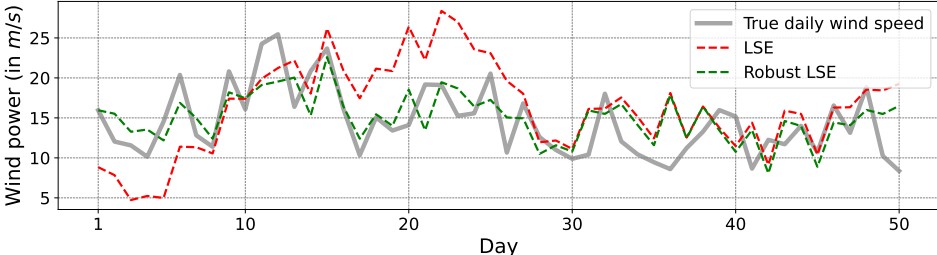

Figure 1: Daily wind speed predictions for a 50-day period. The training sample size is 30 for both LSE (red) and robust LSE (green). The predictions for the next 50 days are compared to the actual wind speed (solid gray line).

We replicated the experiments across 20 different datasets with varying training data sizes (i.e., increasing $T$) and recorded the root mean squared errors (RMSE) as a measure of system identification (i.e., prediction) errors. Figure 2 shows that our approach achieves significant improvement over the standard LSE.

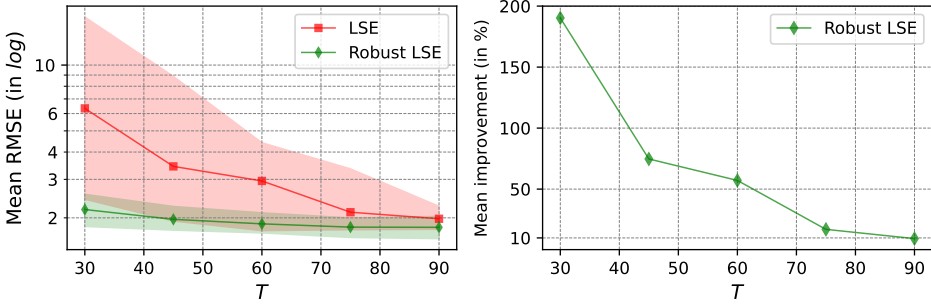

Figure 2: Mean wind speed prediction errors over 20 test datasets: mean RMSE (solid lines) on a log scale, with the 10th and 90th percentiles represented by filled areas (left) and mean percentage improvement of the robust LSE over the standard LSE (right)

## 5.2 LEARNING DYNAMICAL SYSTEMS

Instead of focusing on particular dynamical system examples (in the upcoming online control experiments, we will consider standard examples from the literature), we randomly generated five sets of 500 synthetic systems $\boldsymbol{\theta}^\star = [\boldsymbol{A}^\star \ \boldsymbol{B}^\star] \in \mathbb{R}^{5 \times 10}$, each set having the same spectral radius $\rho(\boldsymbol{A}^\star)$ ranging from 0.1 to 1.0. We compared the system identification errors of the robust LSE and the standard LSE as we collected more samples (i.e., increasing $T$) over time. We observed that the smaller $\rho(\boldsymbol{A}^\star)$ is, the greater performance improvement the robust LSE achieves over the standard LSE. Figure 3 shows the mean system identification errors in the operator norm when $\rho(\boldsymbol{A}^\star) = 1.0$.

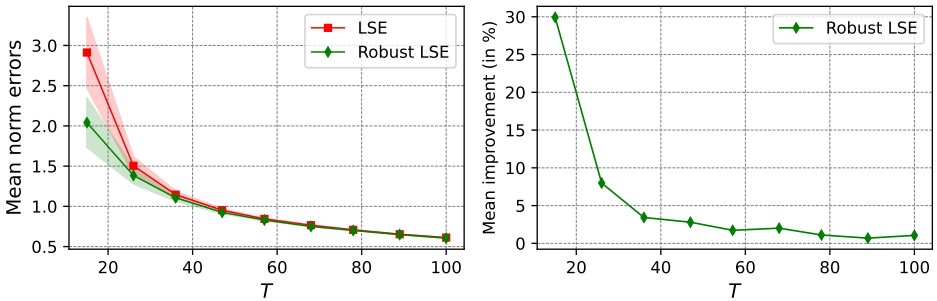

Figure 3: Mean system identification errors over 500 synthetic systems with $\rho(\boldsymbol{A}^\star) = 1.0$: mean errors (solid lines) in the operator norm, with the 10th and 90th percentiles represented by filled areas (left) and mean percentage improvement of the robust LSE over the standard LSE (right)

In the supplementary material F, we tested the stability of the synthesized controller based on the estimated systems in a marginally stable system. We also conducted additional experiments, including one on a high-dimensional system, and another comparing our method with the Wasserstein model.

## 5.3 ONLINE LINEAR QUADRATIC CONTROL

To showcase how our robust LSE can be used to design reliable control systems, we performed online LQ control tasks using standard examples in the literature: i) the longitudinal flight control of Boeing 747 from Ishihara et al. (1992), ii) a marginally unstable Laplacian system from Dean et al. (2020), and iii) UAV in a 2D plane from Zhao et al. (2021). We considered several online LQ algorithms proposed in recent years: 1) **OFULQ** from Abbasi-Yadkori & Szepesvári (2011), 2) **STABL** from Lale et al. (2022), 3) **ARBMLE** from Mete et al. (2022). Broadly speaking, these algorithms conduct two main tasks: identifying the system and deriving the best control input. In particular, OFULQ and STABL utilize the standard LSE for their system identification task. Hence, we can replace the standard LSE with the robust LSE which we referred to as 4) **R-OFULQ** and 5) **R-STABL**.

For each of the algorithms 1)-5), we conducted 500 simulations over a time horizon of $T = 1000$, recording the mean regrets. Due to space limitations, we present only the results for the Boeing 747 example in Figure 4—see the plots for other examples included in the supplementary material G. In Figure 4, we show only the robust algorithms 4) and 5) for $q = 1$ in the SDP (7) to maintain clarity of the plot. As mentioned, however, our online control algorithms can be adjusted by selecting different values of the parameter $q$, corresponding to the Schatten $q$-norm in the SDP (7). A detailed discussion on choosing the parameter $q$ is provided in the supplementary material H.

Every algorithm presented in our experiments requires several parameters. We adopted the parameter setups suggested by the corresponding papers. However, we acknowledge that their setups are not identical to each other. For example, some papers start recording regret after $t = 50$, while others assume a tight upper bound on $\|\boldsymbol{\theta}^\star - \overline{\boldsymbol{\theta}}_t\|$ is available at each time step $t$. Irrespective of the choice of the Schatten norm parameter $q$, our algorithms offer significant advantages over other benchmark algorithms. The results demonstrate not only that the robust LSE can be utilized for various online control algorithms, but also that optimizing the regularization parameter in real-time (i.e., with respect to $T$) for both the robust LSE and the regularized LSE is indeed advantageous.

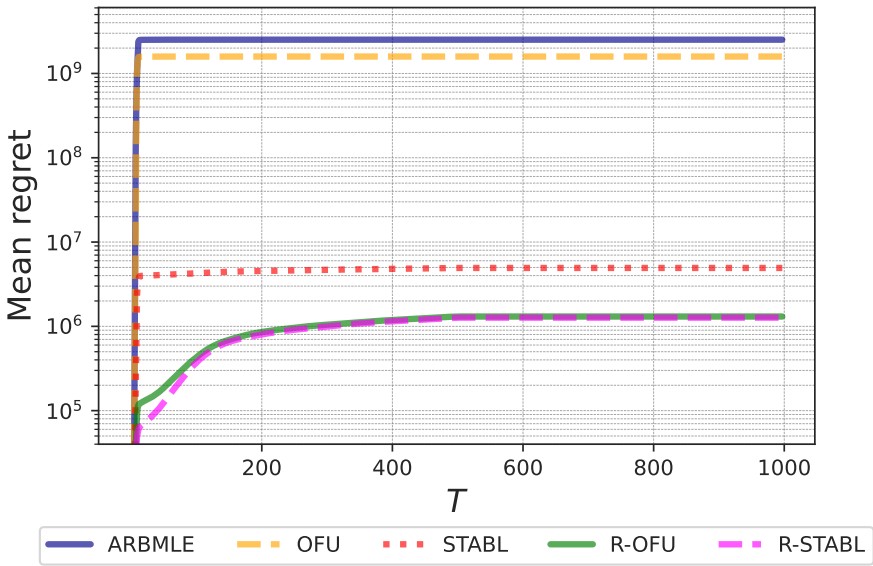

Figure 4: Mean regret over 500 replications: **i)** Boeing 747

## 6 CONCLUDING REMARKS

We present a robust framework for system identification by leveraging robust optimization to immunize standard LSE against small-sample estimation errors and model misspecifications. We derive non-asymptotic guarantees on system identification errors by analyzing the concentration of a single-sample trajectory. Notably, robustifying the estimation achieves a near-optimal error rate and demonstrates substantial empirical improvements. While our analysis is based on a single trajectory, our framework can be straightforwardly applied to a simpler setting where multiple trajectories are available.

Our proposed formulation constitutes a simple semidefinite program, which can be efficiently implemented using standard off-the-shelf solvers. In the special case where the $\infty$-norm is used in the uncertainty set, the formulation reduces to an efficiently solvable quadratic program. The experimental results on the real-world wind prediction problem highlight the significant advantages of our robust model, achieving unprecedented performance. When deployed in online LQ control algorithms, the robust system estimates yield substantially lower regret than standard LSE, further demonstrating the practical benefits of our approach.

Our current work focuses on fully observable systems, which presents a key limitation. Future research will aim to develop a robust optimization framework for identifying partially observable systems with performance guarantees. Additionally, while the complexity of our SDP formulation in (8) is independent of the number of samples, it becomes impractical for high-dimensional systems, as solving SDPs generally scales poorly with dimensionality. For large-scale system identification, we recommend setting $q = 1$ in (8) to reduce the SDP to a quadratic program, though this approach sacrifices the flexibility of selecting the Schatten norm parameter.

### ACKNOWLEDGMENTS

This work is supported by the National Science Foundation grants 2343869 and 2404413.

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

## SUPPLEMENTARY MATERIAL

## A  PROOF OF THEOREM 1

*Proof.* Dualizing the inner maximization problem with the constraint $\|\boldsymbol{\Omega} - \widehat{\boldsymbol{\Omega}}_T\|_p \leq \epsilon$ given by our uncertainty set, we have

$$\max_{\boldsymbol{\Omega} \succeq \mathbf{0}} \min_{\lambda \geq 0} \operatorname{tr}(\boldsymbol{G}(\boldsymbol{\theta})\boldsymbol{\Omega}) + \lambda\epsilon - \lambda\|\boldsymbol{\Omega} - \widehat{\boldsymbol{\Omega}}_T\|_p$$

$$= \max_{\boldsymbol{\Omega} \succeq \mathbf{0}} \min_{\lambda \geq 0} \operatorname{tr}(\boldsymbol{G}(\boldsymbol{\theta})\boldsymbol{\Omega}) + \lambda\epsilon - \max_{\|\boldsymbol{\Gamma}\|_q \leq \lambda} \operatorname{tr}\left(\boldsymbol{\Gamma}\left(\boldsymbol{\Omega} - \widehat{\boldsymbol{\Omega}}_T\right)\right) \tag{13}$$

$$= \max_{\boldsymbol{\Omega} \succeq \mathbf{0}} \min_{\lambda \geq 0} \operatorname{tr}(\boldsymbol{G}(\boldsymbol{\theta})\boldsymbol{\Omega}) + \lambda\epsilon + \min_{\|\boldsymbol{\Gamma}\|_q \leq \lambda} \operatorname{tr}\left(\boldsymbol{\Gamma}\left(\widehat{\boldsymbol{\Omega}}_T - \boldsymbol{\Omega}\right)\right) \tag{14}$$

$$= \min_{\substack{\lambda \geq 0, \\ \|\boldsymbol{\Gamma}\|_q \leq \lambda}} \operatorname{tr}\left(\boldsymbol{\Gamma}\widehat{\boldsymbol{\Omega}}_T\right) + \lambda\epsilon + \max_{\boldsymbol{\Omega} \succeq \mathbf{0}} \operatorname{tr}((\boldsymbol{G}(\boldsymbol{\theta}) - \boldsymbol{\Gamma})\boldsymbol{\Omega}) \tag{15}$$

$$= \min_{\substack{\lambda \geq 0, \\ \|\boldsymbol{\Gamma}\|_q \leq \lambda}} \operatorname{tr}\left(\boldsymbol{\Gamma}\widehat{\boldsymbol{\Omega}}_T\right) + \lambda\epsilon \;\; \text{s.t.} \; \boldsymbol{\Gamma} \succeq \begin{bmatrix} \mathbb{I}_n & -\boldsymbol{\theta} \\ -\boldsymbol{\theta}^\top & \boldsymbol{\theta}^\top\boldsymbol{\theta} \end{bmatrix} \tag{16}$$

$$= \min_{\substack{\lambda \geq 0, \\ \|\boldsymbol{\Gamma}\|_q \leq \lambda, \\ \boldsymbol{H} \succeq \mathbf{0}}} \operatorname{tr}\left(\boldsymbol{\Gamma}\widehat{\boldsymbol{\Omega}}_T\right) + \lambda\epsilon \;\; \text{s.t.} \; \boldsymbol{\Gamma} \succeq \begin{bmatrix} \mathbb{I}_n & -\boldsymbol{\theta} \\ -\boldsymbol{\theta}^\top & \boldsymbol{H} \end{bmatrix}, \; \begin{bmatrix} \mathbb{I}_n & \boldsymbol{\theta} \\ \boldsymbol{\theta}^\top & \boldsymbol{H} \end{bmatrix} \succeq \mathbf{0}. \tag{17}$$

In the first equality (13), we use the definition of the dual norm for $\lambda\|\boldsymbol{\Omega} - \widehat{\boldsymbol{\Omega}}_T\|_p$. As in the second equality (14), we can convert the maximization to a minimization since $\max f(\cdot) = -\min -f(\cdot)$. The third equality (15) exploits strong duality by following the standard results of the convex analysis (see Theorem 1, Chapter 8 in Luenberger (1997)). The feasible set of $(\lambda, \boldsymbol{\Gamma})$ defined in (14) is a convex set, and the objective function of the inner minimization problem is convex in $(\lambda, \boldsymbol{\Gamma})$. Furthermore, we can show the existence of an interior point in the feasible set, that is, there always exists some $\boldsymbol{\Gamma}$ such that the following strict inequality holds: $\|\boldsymbol{\Gamma}\|_q < \lambda$ for any $\lambda > 0$. Hence, strong duality holds. Then, the maximization over $\boldsymbol{\Omega}$ in (15) leads to a restriction of the feasible set which is given by the constraint in (16). In other words, $(\boldsymbol{G}(\boldsymbol{\theta}) - \boldsymbol{\Gamma})$ in (15) needs to be negative semidefinite. In the last equality, we linearize the quadratic term $\boldsymbol{\theta}^\top\boldsymbol{\theta}$ by following Lemma 4 in Mittal et al. (2020). Then, we can combine the minimization in (17) with the minimization over $\boldsymbol{\theta}$ in (7). Finally, reversing the epigraphic reformulation $\|\boldsymbol{\Gamma}\|_q \leq \lambda$ in the equality (17) yields the problem formulation (7), which is a semidefinite program. $\qquad\square$

## B  PROOF OF COROLLARY 1

*Proof.* By reversing the epigraphic reformulation $\|\boldsymbol{\Gamma}\|_q \leq \lambda$ in (16), we have

$$\min_{\boldsymbol{\Gamma}, \boldsymbol{\theta}} \operatorname{tr}\left(\boldsymbol{\Gamma}\widehat{\boldsymbol{\Omega}}_T\right) + \epsilon\|\boldsymbol{\Gamma}\|_q \;\; \text{s.t.} \; \boldsymbol{\Gamma} \succeq \underbrace{\begin{bmatrix} \mathbb{I}_n & -\boldsymbol{\theta} \\ -\boldsymbol{\theta}^\top & \boldsymbol{\theta}^\top\boldsymbol{\theta} \end{bmatrix}}_{=\boldsymbol{G}(\boldsymbol{\theta})}. \tag{18}$$

Suppose that $\boldsymbol{A}, \boldsymbol{B}, \boldsymbol{C} \succeq \mathbf{0}$ and $\boldsymbol{A} \succeq \boldsymbol{B}$. Then, the following is true: $\operatorname{tr}(\boldsymbol{A}\boldsymbol{C}) \geq \operatorname{tr}(\boldsymbol{B}\boldsymbol{C})$. Recall that positive semidefinite inequality $\succeq$ implies ordering on matrices known as Loewner's ordering. One property of the Loewner's ordering is that $\boldsymbol{A} \succeq \boldsymbol{B} \Rightarrow \sigma_i(\boldsymbol{A}) \geq \sigma_i(\boldsymbol{B})$ for all $i$ where $\sigma_i(\cdot)$ denotes the $i$-th singular value of the corresponding matrix (note that the converse is not necessarily true). Also, by definition, the Schatten $q$-norm is equivalent to the $\ell^q$-norm of the vector of singular values, i.e., $\|\boldsymbol{A}\|_q = \|[\sigma_1(\boldsymbol{A}), \ldots, \sigma_n(\boldsymbol{A})]^\top\|_q = (\sum_{i=1}^n |\sigma_i(\boldsymbol{A})|^q)^{1/q}$. Using these facts, we can conclude that $\boldsymbol{\Gamma} = \boldsymbol{G}(\boldsymbol{\theta})$ holds when $\boldsymbol{\Gamma}$ and $\boldsymbol{\theta}$ are the minimizer of the problem (18). Hence, the problem (18) is equivalent to (8). $\qquad\square$

## C  PROOF OF PROPOSITION 1

*Proof.* Proving that the guarantee (10) holds amounts to showing that the distance between $\boldsymbol{\Omega}_T^\star$ and $\widehat{\boldsymbol{\Omega}}_T$ is small w.h.p.: $\|\boldsymbol{\Omega}_T^\star - \widehat{\boldsymbol{\Omega}}_T\|_p \leq \epsilon(\delta)$ w.p. at least $1 - \delta$. Here, we derive the upper bound $\epsilon(\delta)$ for $p = \infty$, i.e., the case where the norm in (11) defined by the Schatten $\infty$-norm (equivalently, operator norm $\|\cdot\| = \|\cdot\|_\infty$). Due to the equivalence of norms, it is easy to show similar bounds for any $p \geq 1$ with different dimensional factors.

Note that $\widehat{\boldsymbol{\Omega}}_T$ can be explicitly expressed as follows:

$$\widehat{\boldsymbol{\Omega}}_T = \frac{1}{T} \sum_{t=0}^{T-1} \begin{bmatrix} \boldsymbol{x}_{t+1} \\ \boldsymbol{x}_t \\ \boldsymbol{u}_t \end{bmatrix} \begin{bmatrix} \boldsymbol{x}_{t+1} \\ \boldsymbol{x}_t \\ \boldsymbol{u}_t \end{bmatrix}^\top \tag{19}$$

$$= \frac{1}{T} \sum_{t=0}^{T-1} \begin{bmatrix} \boldsymbol{x}_{t+1}\boldsymbol{x}_{t+1}^\top & (\boldsymbol{A}^\star\boldsymbol{x}_t + \boldsymbol{B}^\star\boldsymbol{u}_t + \boldsymbol{w}_t)\boldsymbol{x}_t^\top & (\boldsymbol{A}^\star\boldsymbol{x}_t + \boldsymbol{B}^\star\boldsymbol{u}_t + \boldsymbol{w}_t)\boldsymbol{u}_t^\top \\ \boldsymbol{x}_t (\boldsymbol{A}^\star\boldsymbol{x}_t + \boldsymbol{B}^\star\boldsymbol{u}_t + \boldsymbol{w}_t)^\top & \boldsymbol{x}_t\boldsymbol{x}_t^\top & \boldsymbol{x}_t\boldsymbol{u}_t^\top \\ \boldsymbol{u}_t (\boldsymbol{A}^\star\boldsymbol{x}_t + \boldsymbol{B}^\star\boldsymbol{u}_t + \boldsymbol{w}_t)^\top & \boldsymbol{u}_t\boldsymbol{x}_t^\top & \boldsymbol{u}_t\boldsymbol{u}_t^\top \end{bmatrix}. \tag{20}$$

Similarly, $\boldsymbol{\Omega}_T^\star$ is expectation of (20), i.e., $\boldsymbol{\Omega}_T^\star = \mathbb{E}[\widehat{\boldsymbol{\Omega}}_T]$. Hence, using (20), we can establish the following inequalities:

$$\|\boldsymbol{\Omega}_T^\star - \widehat{\boldsymbol{\Omega}}_T\| \leq 2\left(1 + \|\boldsymbol{A}^\star\|\right) \underbrace{\frac{1}{T} \left\| \mathbb{E}\left[\sum_{t=0}^{T-1} \boldsymbol{x}_t\boldsymbol{x}_t^\top\right] - \sum_{t=0}^{T-1} \boldsymbol{x}_t\boldsymbol{x}_t^\top \right\|}_{(a)}$$

$$+ 2 \underbrace{\frac{1}{T} \left\| \mathbb{E}\left[\sum_{t=0}^{T-1} \boldsymbol{w}_t\boldsymbol{x}_t^\top\right] - \sum_{t=0}^{T-1} \boldsymbol{w}_t\boldsymbol{x}_t^\top \right\|}_{(b)} + 2\left(1 + \|\boldsymbol{A}^\star\| + \|\boldsymbol{B}^\star\|\right) \underbrace{\frac{1}{T} \left\| \mathbb{E}\left[\sum_{t=0}^{T-1} \boldsymbol{u}_t\boldsymbol{x}_t^\top\right] - \sum_{t=0}^{T-1} \boldsymbol{u}_t\boldsymbol{x}_t^\top \right\|}_{(c)}$$

$$+ \left(1 + 2\|\boldsymbol{B}^\star\|\right) \underbrace{\frac{1}{T} \left\| \mathbb{E}\left[\sum_{t=0}^{T-1} \boldsymbol{u}_t\boldsymbol{u}_t^\top\right] - \sum_{t=0}^{T-1} \boldsymbol{u}_t\boldsymbol{u}_t^\top \right\|}_{(d)} + 2 \underbrace{\frac{1}{T} \left\| \mathbb{E}\left[\sum_{t=0}^{T-1} \boldsymbol{w}_t\boldsymbol{u}_t^\top\right] - \sum_{t=0}^{T-1} \boldsymbol{w}_t\boldsymbol{u}_t^\top \right\|}_{(e)}.$$

Our goal is to bound each of the terms (a)-(e), and then combine the results to complete the proof.

(a):

Notice that we analyze the difference between the Gram matrix and its expectation with factor $(1/T)$. Similar results are discussed in Jedra & Proutiere (2020). First, we introduce the preparatory result.

Suppose $\rho(\boldsymbol{A}) < 1$ for a matrix $\boldsymbol{A} \in \mathbb{R}^{n \times n}$. Consider a $t \times t$ block Toeplitz matrix

$$\boldsymbol{H}_t = \begin{bmatrix} \mathbb{I}_n & \boldsymbol{0} & \boldsymbol{0} & \boldsymbol{0} \\ \boldsymbol{A} & \mathbb{I}_n & \boldsymbol{0} & \boldsymbol{0} \\ \vdots & \vdots & \ddots & \boldsymbol{0} \\ \boldsymbol{A}^{t-1} & \boldsymbol{A}^{t-2} & \cdots & \mathbb{I}_n \end{bmatrix} \in \mathbb{R}^{nt \times nt}. \tag{21}$$

Then, for any $t \geq 1$, there exists a finite constant $\mathcal{J}(\boldsymbol{A}) > 0$ that only depends on $\boldsymbol{A}$ such that

$$\|\boldsymbol{H}_t\| \leq \mathcal{J}(\boldsymbol{A}) := \sum_{s=0}^{+\infty} \|\boldsymbol{A}^s\|, \tag{22}$$

where $\mathcal{J}(\boldsymbol{A})$ is specifically the limit of a matrix power series $\sum_{s=0}^{t} \|\boldsymbol{A}^s\|$.

Jedra & Proutiere (2020) analyze the sample complexity of the unregularized LSE where an unknown system is uncontrolled. (i.e., identifying only $\boldsymbol{A}^\star$). We can derive a similar result to Lemma 2 in their work.

Under an i.i.d. sub-Gaussian exploration noise, our dynamic system can be written as $\boldsymbol{x}_{t+1} = \boldsymbol{A}^\star \boldsymbol{x}_t + \boldsymbol{\eta}_t$ where $\boldsymbol{\eta}_t$ is a zero mean noise with a covariance matrix $\boldsymbol{\Sigma}_\eta := \mathbb{E}[\boldsymbol{\eta}_t \boldsymbol{\eta}_t^\top] = \sigma_u^2 \boldsymbol{B}^\star \boldsymbol{B}^{\star\top} + \boldsymbol{\Sigma}_w$. Then, we can define vectorized states of the system up to time $T$:

$$
\begin{bmatrix} \boldsymbol{x}_1 \\ \vdots \\ \boldsymbol{x}_T \end{bmatrix} = \mathbf{H}_T \boldsymbol{C}_\eta^{1/2} \boldsymbol{\xi} \in \mathbb{R}^{nT} \text{ where } \boldsymbol{C}_\eta = \mathbb{E}\left[ \begin{bmatrix} \boldsymbol{\eta}_0 \\ \vdots \\ \boldsymbol{\eta}_{T-1} \end{bmatrix} \begin{bmatrix} \boldsymbol{\eta}_0 \\ \vdots \\ \boldsymbol{\eta}_{T-1} \end{bmatrix}^\top \right] = \begin{bmatrix} \boldsymbol{\Sigma}_\eta & \mathbf{0} & \mathbf{0} \\ \mathbf{0} & \ddots & \mathbf{0} \\ \mathbf{0} & \mathbf{0} & \boldsymbol{\Sigma}_\eta \end{bmatrix} \in \mathbb{S}_+^{nT}
$$

$$
\text{and } \boldsymbol{\xi} = \begin{bmatrix} \boldsymbol{\xi}_0 \\ \vdots \\ \boldsymbol{\xi}_{T-1} \end{bmatrix} \in \mathbb{R}^{nT} \text{ is isotropic, i.e., } \mathbb{E}[\boldsymbol{\xi}\boldsymbol{\xi}^\top] = \mathbb{I}_{nT} \tag{23}
$$

To simplify the notation, let us define the reciprocal of the square root matrix of the expected Gram matrix as follows:

$$
\boldsymbol{M} := \left( \sum_{t=0}^{T-1} \boldsymbol{\Gamma}_t(\boldsymbol{\theta}^\star) \right)^{-1/2} = \left( \sum_{t=0}^{T-1} \sum_{s=0}^{t} (\boldsymbol{A}^{\star\top})^s (\sigma_u^2 \boldsymbol{B}^\star \boldsymbol{B}^{\star\top} + \boldsymbol{\Sigma}_w)(\boldsymbol{A}^\star)^s \right)^{-1/2}.
$$

Then, we can establish the following equalities:

$$
\left\| \boldsymbol{M}^\top \sum_{t=0}^{T-1} \boldsymbol{x}_t \boldsymbol{x}_t^\top \boldsymbol{M} - \mathbb{I}_n \right\| = \sup_{\|\boldsymbol{u}\|_2 \le 1} \left| \boldsymbol{u}^\top \left( \boldsymbol{M}^\top \sum_{t=0}^{T-1} \boldsymbol{x}_t \boldsymbol{x}_t^\top \boldsymbol{M} - \mathbb{I}_n \right) \boldsymbol{u} \right| \tag{24}
$$

$$
= \sup_{\|\boldsymbol{u}\|_2 \le 1} \left| \left\| \sum_{t=0}^{T-1} \boldsymbol{x}_t^\top \boldsymbol{M} \boldsymbol{u} \right\|_2^2 - \mathbb{E}\left[ \left\| \Sigma_{t=0}^{T-1} \boldsymbol{x}_t^\top \boldsymbol{M} \boldsymbol{u} \right\|_2^2 \right] \right| \tag{25}
$$

$$
= \sup_{\|\boldsymbol{u}\|_2 \le 1} \left| \left\| \boldsymbol{\Sigma}_{\boldsymbol{M}\boldsymbol{u}}^\top \mathbf{H}_T \mathbf{C}_\eta^{1/2} \boldsymbol{\xi} \right\|_2^2 - \mathbb{E}\left[ \left\| \boldsymbol{\Sigma}_{\boldsymbol{M}\boldsymbol{u}}^\top \mathbf{H}_T \mathbf{C}_\eta^{1/2} \boldsymbol{\xi} \right\|_2^2 \right] \right| \tag{26}
$$

$$
= \sup_{\|\boldsymbol{u}\|_2 \le 1} \left| \left\| \boldsymbol{\Sigma}_{\boldsymbol{M}\boldsymbol{u}}^\top \mathbf{H}_T \mathbf{C}_\eta^{1/2} \boldsymbol{\xi} \right\|_2^2 - \left\| \boldsymbol{\Sigma}_{\boldsymbol{M}\boldsymbol{u}}^\top \mathbf{H}_T \mathbf{C}_\eta^{1/2} \right\|_F^2 \right|, \tag{27}
$$

where $\boldsymbol{\Sigma}_{\boldsymbol{M}\boldsymbol{u}} = \begin{bmatrix} \boldsymbol{M}\boldsymbol{u} & \mathbf{0} & \mathbf{0} \\ \mathbf{0} & \ddots & \mathbf{0} \\ \mathbf{0} & \mathbf{0} & \boldsymbol{M}\boldsymbol{u} \end{bmatrix} \in \mathbb{R}^{nT \times T}$ in (26) is a block diagonal matrix.

The first equality (24) is the variational form of the operator norm. In the last equality (27), we use the fact that $\mathbb{E}[\|\boldsymbol{D}\boldsymbol{\xi}\|_2^2] = \mathrm{tr}(\boldsymbol{D}^\top \boldsymbol{D} \mathbb{E}[\boldsymbol{\xi}\boldsymbol{\xi}^\top]) = \|\boldsymbol{D}\|_F^2$ for an isotropic vector $\boldsymbol{\xi}$. The objective function in (27) can be written as $\left| |\boldsymbol{\xi}^\top \boldsymbol{W} \boldsymbol{\xi}| - |\mathbb{E}[\boldsymbol{\xi}^\top \boldsymbol{W} \boldsymbol{\xi}]| \right|$ where $(\boldsymbol{\xi}^\top \boldsymbol{W} \boldsymbol{\xi})_{\boldsymbol{W} \in \mathcal{W}}$ indexed by a set of matrices $\mathcal{W}$ is referred to as a chaos process.

We omit the remaining steps since they are identical to the proof of Lemma 2 in Jedra & Proutiere (2020) once we recognize that (27) is the supremum of a chaos process. The main idea for the remaining steps is that the Hanson-Wright inequality (Hanson & Wright, 1971) provides the concentration bound on (27) when $\boldsymbol{u}$ is fixed. Then, we can use the $\epsilon$-net argument, i.e., discretizing the feasible region $\mathcal{U} = \{\boldsymbol{u} : \|\boldsymbol{u}\|_2 \le 1\}$ and combining the bounds for all $\boldsymbol{u} \in \mathcal{U}(\epsilon)$ where $\mathcal{U}(\epsilon)$ is an $\epsilon$-net of $\mathcal{U}$. Following this idea, for $\delta \in (0, 1]$, we have

$$
\Pr\left[ \frac{1}{T} \left\| \mathbb{E}\left[ \sum_{t=0}^{T-1} \boldsymbol{x}_t \boldsymbol{x}_t^\top \right] - \sum_{t=0}^{T-1} \boldsymbol{x}_t \boldsymbol{x}_t^\top \right\| \le \epsilon_{(a)}(\delta) \right] \ge 1 - \delta, \text{ where}
$$

$$
\epsilon_{(a)}(\delta) = \sigma_w^2 \max\left\{ \frac{\sqrt{\|\boldsymbol{M}^{-1}\| \|\boldsymbol{H}_T\|^2 \|\mathbf{C}_\eta\| \left( \log\left(\frac{2}{\delta}\right) + c_2 n \right)}}{\sqrt{c_1} T}, \frac{\|\boldsymbol{H}_T\|^2 \|\mathbf{C}_\eta\| \left( \log\left(\frac{2}{\delta}\right) + c_2 n \right)}{c_1 T} \right\}. \tag{28}
$$

Note that $\|\boldsymbol{H}_T\|$ in (28) can be further bounded by some finite constant $\mathcal{J}(\boldsymbol{A}^\star)$ due to the preparatory result (22). However, we have not made the explicit dependence of $\epsilon_{(a)}(\delta)$ in terms of $T$ yet as $\|\boldsymbol{M}^{-1}\|$ in (28) grows with $T$. We defer the discussion to where the bounds on (b) and (c) are established since the same issue arises.

(b) and (c):

The same technique is applied to (b) and (c). Hence, we only show the derivation for (b). Note that since the noise term $\boldsymbol{w}_t$ is independent of $\boldsymbol{x}_t$, the expectation in (b) is a zero matrix. Hence, we only need to analyze $(1/T)\|\sum_{t=0}^{T-1} \boldsymbol{w}_t \boldsymbol{x}_t^\top\|$. Assuming $\sum_{t=0}^{T-1} \boldsymbol{x}_t \boldsymbol{x}_t^\top$ is invertible (at the moment), we can break (b) into the product of two terms as follows:

$$
\frac{1}{T}\Big\|\sum_{t=0}^{T-1} \boldsymbol{w}_t \boldsymbol{x}_t^\top\Big\| = \frac{1}{T}\Big\|\Big(\sum_{t=0}^{T-1} \boldsymbol{w}_t \boldsymbol{x}_t^\top\Big)\Big(\sum_{t=0}^{T-1} \boldsymbol{x}_t \boldsymbol{x}_t^\top\Big)^{-1/2}\Big(\sum_{t=0}^{T-1} \boldsymbol{x}_t \boldsymbol{x}_t^\top\Big)^{1/2}\Big\|
$$

$$
\leq \frac{1}{T}\underbrace{\Big\|\Big(\sum_{t=0}^{T-1} \boldsymbol{w}_t \boldsymbol{x}_t^\top\Big)\Big(\sum_{t=0}^{T-1} \boldsymbol{x}_t \boldsymbol{x}_t^\top\Big)^{-1/2}\Big\|}_{\text{self-normalized martingale}}\underbrace{\Big\|\Big(\sum_{t=0}^{T-1} \boldsymbol{x}_t \boldsymbol{x}_t^\top\Big)^{1/2}\Big\|}_{\text{persistent excitation term}}. \quad (29)
$$

As denoted, the stochastic process in (29) is referred to as the self-normalized martingale whose non-asymptotic bounds are already analyzed in Abbasi-Yadkori & Szepesvári (2011). Hence, we can invoke the following results to obtain the bound on the self-normalized term.

Suppose that $\boldsymbol{V}_T = \sum_{t=0}^{T-1} \boldsymbol{x}_t \boldsymbol{x}_t^\top + \boldsymbol{V}$ where $\boldsymbol{V} = c\lfloor T/2\rfloor\,\boldsymbol{\Gamma}_1(\boldsymbol{\theta}^\star)$ is a positive definite matrix with a universal constant $c > 0$, ensuring the invertibility of $\boldsymbol{V}_T$. Then, for $\delta \in (0,1]$, we have

$$
\mathbb{P}\Bigg[\Big\|\Big(\sum_{t=0}^{T-1} \boldsymbol{w}_t \boldsymbol{x}_t^\top\Big)\Big(\sum_{t=0}^{T-1} \boldsymbol{x}_t \boldsymbol{x}_t^\top\Big)^{-1/2}\Big\| \leq 4\sqrt{\|\boldsymbol{\Sigma}_w\|\log\Bigg(\sqrt{\frac{\det(\boldsymbol{V}_T)}{\det(\boldsymbol{V})}}\cdot\frac{5^n}{\delta}\Bigg)}\Bigg] \geq 1-\delta \quad (30)
$$

$$
\text{as long as } T \geq \mathcal{O}\left(n\log\left(\frac{n}{\delta}\right) + \log\left(\frac{\det \boldsymbol{\Gamma}_T(\boldsymbol{\theta}^\star)}{\det \boldsymbol{\Gamma}_1(\boldsymbol{\theta}^\star)}\right)\right). \quad (31)
$$

Note that $\boldsymbol{V}_T$ in (30) is the only term that depends on $T$ and it increases at most logarithmically as $T$ grows. We make a few comments before proceeding: i) the bound (30) has to be probabilistic since the invertibility (i.e., positive definiteness) of $\sum_{t=0}^{T-1} \boldsymbol{x}_t \boldsymbol{x}_t^\top$ cannot be guaranteed deterministically; ii) the lower bound on $T$ in (31), i.e., the minimum number of samples that ensures the invertibility of $\sum_{t=0}^{T-1} \boldsymbol{x}_t \boldsymbol{x}_t^\top$ w.h.p., is called the burn-in time. Here, we use the big-O notation for the burn-in time only because we want to streamline the exposition. We make the quantity explicit in the proof of Theorem 2 under sufficient conditions.

Subsequently, we derive an upper bound on the persistent excitation term in (29). Note that the term is similar to one in (28). Hence, we can establish the following inequalities:

$$
\Big\|\mathbb{E}\Big[\sum_{t=0}^{T-1} \boldsymbol{x}_t \boldsymbol{x}_t^\top\Big]^{1/2} - \Big(\sum_{t=0}^{T-1} \boldsymbol{x}_t \boldsymbol{x}_t^\top\Big)^{1/2}\Big\| \leq n^{\frac{1}{4}}\sqrt{\Big\|\mathbb{E}\Big[\sum_{t=0}^{T-1} \boldsymbol{x}_t \boldsymbol{x}_t^\top\Big] - \sum_{t=0}^{T-1} \boldsymbol{x}_t \boldsymbol{x}_t^\top\Big\|} \leq n^{\frac{1}{4}}\sqrt{T\cdot\epsilon_{(a)}(\delta)}
$$
$$(32)$$

w.p. at least $1-\delta$.

In the first inequality, we use the following fact: $\|\boldsymbol{A}^{1/2} - \boldsymbol{B}^{1/2}\| \leq \sqrt{\|\boldsymbol{A}-\boldsymbol{B}\|_F} \leq n^{\frac{1}{4}}\sqrt{\|\boldsymbol{A}-\boldsymbol{B}\|}$ for any $\boldsymbol{A}, \boldsymbol{B} \in \mathbb{S}_+^n$. The second inequality follows from (28). By the reverse triangle inequality, we can further derive the following upper bound on the persistent excitation term:

$$
\Big\|\Big(\sum_{t=0}^{T-1} \boldsymbol{x}_t \boldsymbol{x}_t^\top\Big)^{1/2}\Big\| \leq \underbrace{\Big\|\mathbb{E}\Big[\sum_{t=0}^{T-1} \boldsymbol{x}_t \boldsymbol{x}_t^\top\Big]^{1/2}\Big\|}_{=\boldsymbol{M}^{-1}} + n^{\frac{1}{4}}\sqrt{T\cdot\epsilon_{(a)}(\delta)}. \quad (33)
$$

Recall that we have not addressed the term $\|\boldsymbol{M}^{-1}\|$ in $\epsilon_{(a)}(\delta)$. In fact, the term $\|\mathbb{E}[\sum_{t=0}^{T-1} \boldsymbol{x}_t \boldsymbol{x}_t^\top]^{1/2}\|$ in (33) is equivalent to $\|\boldsymbol{M}^{-1}\|$ as denoted above. Using the definition of the expected Gram matrix

(9), we obtain the following inequalities:

$$
\left\| \mathbb{E}\left[ \sum_{t=0}^{T-1} \boldsymbol{x}_t \boldsymbol{x}_t^\top \right]^{1/2} \right\| = \left\| \mathbb{E}\left[ \sum_{t=0}^{T-1} \boldsymbol{x}_t \boldsymbol{x}_t^\top \right] \right\|^{1/2} = \left\| \sum_{t=0}^{T} \boldsymbol{\Gamma}_t\left(\theta^\star\right) \right\|^{1/2}
$$

$$
= \left\| \sum_{t=0}^{T} \sum_{s=0}^{t} \left(\boldsymbol{A}^\star\right)^s \left(\sigma_u^2 \boldsymbol{B}^\star \boldsymbol{B}^{\star\top} + \boldsymbol{\Sigma}_w\right) \left(\boldsymbol{A}^{\star\top}\right)^s \right\|^{1/2}
$$

$$
\leq \left\| T \sum_{s=0}^{+\infty} \left(\boldsymbol{A}^\star\right)^s \left(\sigma_u^2 \boldsymbol{B}^\star \boldsymbol{B}^{\star\top} + \boldsymbol{\Sigma}_w\right) \left(\boldsymbol{A}^{\star\top}\right)^s \right\|^{1/2}
$$

$$
\leq \sqrt{T} \left\| \sigma_u^2 \boldsymbol{B}^\star \boldsymbol{B}^{\star\top} + \boldsymbol{\Sigma}_w \right\|^{1/2} \left\| \sum_{s=0}^{+\infty} \left(\boldsymbol{A}^\star\right)^s \right\|
$$

$$
= \sqrt{T} \left\| \sigma_u^2 \boldsymbol{B}^\star \boldsymbol{B}^{\star\top} + \boldsymbol{\Sigma}_w \right\|^{1/2} J\left(\boldsymbol{A}^\star\right) \tag{34}
$$

$$
= \mathcal{O}(\sqrt{T}).
$$

The first equality holds since the expected Gram matrix is positive semidefinite and (34) follows from the preparatory result (22). Here, we emphasize $\|\boldsymbol{M}^{-1}\|$ grows at the rate of $\mathcal{O}(\sqrt{T})$. Hence, combining (30) and (33) with the factor $1/T$ yields that (b) is upper-bounded by $\widetilde{\mathcal{O}}(1/\sqrt{T})$. Moreover, since $\|\boldsymbol{M}^{-1}\| = \mathcal{O}(\sqrt{T})$, we can claim that $\epsilon_{(a)}(\delta)$ in (28) is at most $\mathcal{O}(1/\sqrt{T})$.

(d) and (e):

They can be addressed by the standard concentration inequality for a covariance matrix (see Theorem 6.5 in Wainwright (2019)). For (d), under i.i.d. sub-Gaussian exploration noise, we can claim that there exist universal constants $c_1, c_2, c_3 > 0$ such that

$$
\mathbb{P}\left[ \left\| \mathbb{E}\left[ \sum_{t=1}^{T} \boldsymbol{u}_t \boldsymbol{u}_t^\top \right] - \sum_{t=1}^{T} \boldsymbol{u}_t \boldsymbol{u}_t^\top \right\| \leq \epsilon_{(c)}(\delta) \right] \geq 1 - \delta, \tag{35}
$$

where $\epsilon_{(c)}(\delta) = \sigma_u^2 \cdot c_1 \left(\sqrt{\frac{m}{T}} + \frac{m}{T}\right) + \sigma_u^2 \left(\sqrt{\frac{\log\left(\frac{c_2}{\delta}\right)}{Tc_3}} + \frac{\log\left(\frac{c_2}{\delta}\right)}{Tc_3}\right) = \mathcal{O}(1/\sqrt{T})$. For (e), we can apply the same concentration inequality by defining an augmented random vector $\boldsymbol{v}_t = [\boldsymbol{u}_t^\top\ \boldsymbol{w}_t^\top]^\top$ since

$$
\frac{1}{T} \left\| \mathbb{E}\left[ \sum_{t=1}^{T} \boldsymbol{w}_t \boldsymbol{u}_t^\top \right] - \sum_{t=1}^{T} \boldsymbol{w}_t \boldsymbol{u}_t^\top \right\| \leq \frac{1}{T} \left\| \mathbb{E}\left[ \sum_{t=1}^{T} \boldsymbol{v}_t \boldsymbol{v}_t^\top \right] - \sum_{t=1}^{T} \boldsymbol{v}_t \boldsymbol{v}_t^\top \right\|.
$$

Therefore, there exists universal constants $\bar{c}_1, \bar{c}_2, \bar{c}_3 > 0$ such that

$$
\mathbb{P}\left[ \left\| \mathbb{E}\left[ \sum_{t=1}^{T} \boldsymbol{w}_t \boldsymbol{u}_t^\top \right] - \sum_{t=1}^{T} \boldsymbol{w}_t \boldsymbol{u}_t^\top \right\| \leq \epsilon_{(d)}(\delta) \right] \geq 1 - \delta, \tag{36}
$$

where $\epsilon_{(d)}(\delta) = \max\left(\sigma_u^2, \sigma_w^2\right) \cdot \bar{c}_1 \left(\sqrt{\frac{n+m}{T}} + \frac{n+m}{T}\right) + \max\left(\sigma_u^2, \sigma_w^2\right) \left(\sqrt{\frac{\log\left(\frac{\bar{c}_2}{\delta}\right)}{T\bar{c}_3}} + \frac{\log\left(\frac{\bar{c}_2}{\delta}\right)}{T\bar{c}_3}\right) = \mathcal{O}(1/\sqrt{T})$.

Finally, combining (a)-(e), we have

$$
\epsilon(\delta) = c_1 \left( \frac{\sqrt{\log\left(\frac{2}{\delta}\right) + c_2 n}}{T^{\frac{3}{4}}} + \frac{\left(\log\left(\frac{2}{\delta}\right) + c_2 n\right)}{T} \right) + c_3 \frac{1}{\sqrt{T}} \cdot \sqrt{\log(T) + \log\left(\frac{5n}{\delta}\right)}
$$

$$
+ c_4 \left( \sqrt{\frac{n+m}{T}} + \frac{n+m}{T} + \sqrt{\frac{\log\left(\frac{c_5}{\delta}\right)}{T}} + \frac{\log\left(\frac{c_5}{\delta}\right)}{T} \right). \tag{37}
$$

Here, $c_1, c_2, c_3, c_4$, and $c_5$ are universal constants. Removing the dependence of (37) on variables other than $T$ establishes the guarantee (10). $\qquad\square$

# D  PROOF OF THEOREM 2

*Proof.* The guarantee (10) in Proposition 1 implies that the following holds:

$$\mathbb{P}\left[\min_{\boldsymbol{\theta}} \underbrace{\text{tr}\left(\boldsymbol{G}(\boldsymbol{\theta})\boldsymbol{\Omega}_T^\star\right)}_{=f(\boldsymbol{\theta})} \leq \min_{\boldsymbol{\theta}} \underbrace{\text{tr}\left(\boldsymbol{G}(\boldsymbol{\theta})\widehat{\boldsymbol{\Omega}}_T\right) + \epsilon(\delta)\|\boldsymbol{G}(\boldsymbol{\theta})\|_q}_{=g(\boldsymbol{\theta})}\right] \geq 1 - \delta.$$

Let $f(\boldsymbol{\theta})$ and $g(\boldsymbol{\theta})$ be the objective function of the true and robust LSE problems, respectively. First, we show that $g(\boldsymbol{\theta})$ is an $\alpha$-strongly convex function with high probability (w.h.p.). Following the definition of strong convexity, showing strong convexity amounts to showing that $g(\boldsymbol{\theta})$ can be rewritten as $g(\boldsymbol{\theta}) = g'(\boldsymbol{\theta}) + \alpha\|\boldsymbol{\theta}\|_F^2$ where $g'(\boldsymbol{\theta})$ is a convex function and $\alpha > 0$. Note that $\text{tr}(\boldsymbol{G}(\boldsymbol{\theta})\widehat{\boldsymbol{\Omega}}_T)$ contains the convex quadratic function, i.e., $\text{tr}(1/T\sum_{t=0}^T \boldsymbol{z}_t\boldsymbol{z}_t^\top \boldsymbol{\theta}^\top\boldsymbol{\theta})$. As shown in Tsiamis et al. (2023), under i.i.d. exploration noise, the stochastic process of $\boldsymbol{z}_t = [\boldsymbol{x}_t^\top \ \boldsymbol{u}_t^\top]^\top$ satisfies the block martingale small ball (BMSB) condition with parameters $(k, \widetilde{\boldsymbol{\Gamma}}_{\lfloor k/2\rfloor}, 3/20)$ where parameter $k$ can be set to a positive integer and

$$\widetilde{\boldsymbol{\Gamma}}_{\lfloor k/2\rfloor} = \begin{bmatrix} \boldsymbol{\Gamma}_{\lfloor k/2\rfloor}(\boldsymbol{\theta}^\star) & \boldsymbol{0} \\ \boldsymbol{0} & \sigma_u^2\mathbb{I}_m \end{bmatrix} \text{ is the covariance matrix of } \boldsymbol{z}_{\lfloor k/2\rfloor}.$$

It can be shown that the BMSB condition can guarantee the persistent excitation w.h.p. (see Proposition 2.5 in Simchowitz et al. (2018)). Therefore, by setting $k = 2$, we can establish the following persistent excitation of the stochastic process $\boldsymbol{z}_t$ for $T \geq T(\delta)$ (defined earlier):

$$\mathbb{P}\left[\frac{1}{T}\sum_{t=0}^T \boldsymbol{z}_t\boldsymbol{z}_t^\top \succeq \widehat{\alpha}\mathbb{I}_{(n+m)}\right] \geq 1 - \delta, \text{ where } \widehat{\alpha} = \frac{1}{16}\left(\frac{3}{20}\right)^2\left(\frac{2}{3}\right)\min\{\sigma_w^2, \sigma_u^2\}. \quad (38)$$

Hence, we can claim that for any significance level $\delta \in (0, 1]$, $g(\boldsymbol{\theta})$ is $\widehat{\alpha}$-strongly convex with probability (w.p.) at least $1 - \delta$ when $T$ is sufficiently large. Suppose $g(\boldsymbol{\theta})$ is indeed an $\widehat{\alpha}$-strongly convex function. Then, we can upper-bound the system identification errors as follows:

$$\|\boldsymbol{\theta}^\star - \widehat{\boldsymbol{\theta}}_T\|_F \leq \frac{2}{\widehat{\alpha}}\|\nabla_{\boldsymbol{\theta}}g(\boldsymbol{\theta}^\star)\|_F \leq \frac{2\sqrt{\min\{n, m\}}}{\widehat{\alpha}}\|\nabla_{\boldsymbol{\theta}}g(\boldsymbol{\theta}^\star)\|. \quad (39)$$

The first inequality follows from the properties of strong convexity. The second inequality holds due to the equivalence of norms. To ease the notation, we define the following block matrix notations for $\boldsymbol{\Omega}_T^\star$ and $\widehat{\boldsymbol{\Omega}}_T$:

$$\boldsymbol{\Omega}_T^\star = \begin{bmatrix} \boldsymbol{Q}^\star & \boldsymbol{W}^\star \\ \boldsymbol{W}^{\star\top} & \boldsymbol{E}^\star \end{bmatrix} \text{ and } \widehat{\boldsymbol{\Omega}}_T = \begin{bmatrix} \widehat{\boldsymbol{Q}} & \widehat{\boldsymbol{W}} \\ \widehat{\boldsymbol{W}}^\top & \widehat{\boldsymbol{E}} \end{bmatrix}. \quad (40)$$

Then, we can write the gradient in (39) as $\nabla_{\boldsymbol{\theta}}g(\boldsymbol{\theta}^\star) = -2\widehat{\boldsymbol{W}} + 2\boldsymbol{\theta}^\star\widehat{\boldsymbol{E}}^\top + \epsilon(\delta)\nabla_{\boldsymbol{\theta}}\|\boldsymbol{G}(\boldsymbol{\theta}^\star)\|_q$. Subsequently, we can establish the following inequalities:

$$\|\nabla_{\boldsymbol{\theta}}g(\boldsymbol{\theta}^\star)\| = \left\|-2\widehat{\boldsymbol{W}} + 2\boldsymbol{\theta}^\star\widehat{\boldsymbol{E}}^\top + \epsilon(\delta)\nabla_{\boldsymbol{\theta}}\|\boldsymbol{G}(\boldsymbol{\theta}^\star)\|_q\right\|$$

$$\leq \sup_{\left\|\begin{bmatrix}\Delta\boldsymbol{Q} & \Delta\boldsymbol{W} \\ \Delta\boldsymbol{W}^\top & \Delta\boldsymbol{E}\end{bmatrix}\right\|\leq\epsilon(\delta)} \left\|-2\left(\boldsymbol{W}^\star - \Delta\boldsymbol{W}\right) + 2\boldsymbol{\theta}^\star\left(\boldsymbol{E}^\star - \Delta\boldsymbol{E}\right)^\top + \epsilon(\delta)\nabla_{\boldsymbol{\theta}}\|\boldsymbol{G}(\boldsymbol{\theta}^\star)\|_q\right\| \tag{41}$$

$$= \sup_{\left\|\begin{bmatrix}\Delta\boldsymbol{Q} & \Delta\boldsymbol{W} \\ \Delta\boldsymbol{W}^\top & \Delta\boldsymbol{E}\end{bmatrix}\right\|\leq\epsilon(\delta)} \left\|2\Delta\boldsymbol{W} - 2\boldsymbol{\theta}^\star\Delta\boldsymbol{E}^\top + \epsilon(\delta)\nabla_{\boldsymbol{\theta}}\|\boldsymbol{G}(\boldsymbol{\theta}^\star)\|_q\right\| \tag{42}$$

$$\leq \sup_{\left\|\begin{bmatrix}\Delta\boldsymbol{Q} & \Delta\boldsymbol{W} \\ \Delta\boldsymbol{W}^\top & \Delta\boldsymbol{E}\end{bmatrix}\right\|\leq\epsilon(\delta)} 2\|\Delta\boldsymbol{W}\| + 2\|\boldsymbol{\theta}^\star\|\|\Delta\boldsymbol{E}\| + \epsilon(\delta)\|\nabla_{\boldsymbol{\theta}}\|\boldsymbol{G}(\boldsymbol{\theta}^\star)\|_q\|$$

$$\leq 2\epsilon(\delta) + 2\|\boldsymbol{\theta}^\star\|\epsilon(\delta) + \epsilon(\delta)\|\nabla_{\boldsymbol{\theta}}\|\boldsymbol{G}(\boldsymbol{\theta}^\star)\|_q\|$$

$$= \epsilon(\delta)(2 + 2\|\boldsymbol{\theta}^\star\| + \|\nabla_{\boldsymbol{\theta}}\|\boldsymbol{G}(\boldsymbol{\theta}^\star)\|_q\|) \tag{43}$$

The first inequality (41) holds due to our guarantee in Proposition 1. In the next equality (42), we cancel out the terms $\boldsymbol{W}^\star$ and $\boldsymbol{E}^\star$ using the optimality condition for the true LSE problem, namely, $\nabla_{\boldsymbol{\theta}}f(\boldsymbol{\theta}^\star) = \boldsymbol{0} \Rightarrow \boldsymbol{W}^\star = \boldsymbol{\theta}^\star\boldsymbol{E}^{\star\top}$. Combining (38) and (39) ($\|\nabla_{\boldsymbol{\theta}}g(\boldsymbol{\theta}^\star)\|$ in (39) replaced by (43)) using the union bound yields the claim. □

# E  COMPUTATIONAL TIME

| $T$ | **i)** Boeing 747 | **ii)** Laplacian | **iii)** UAV |
|------|------|------|------|
| 100 | 6.09E-02 | 5.58E-02 | 6.61E-02 |
| 400 | 6.30E-02 | 5.10E-02 | 5.20E-02 |
| 1000 | 4.90E-02 | 6.09E-02 | 5.08E-02 |

Table 1: Mean computational time (in seconds) over 100 replications for solving the example systems in the SDP formulation; as shown here, the computational time is invariant to the sample size $T$.

# F  FURTHER EXPERIMENTS

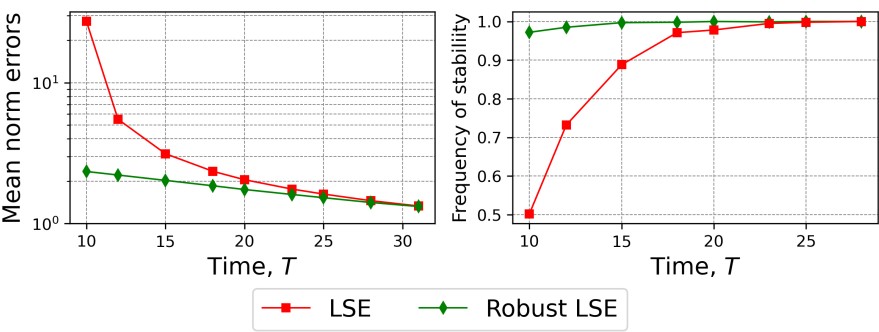

Figure 5: We consider a marginally stable system (i.e., $\rho(\boldsymbol{A}^\star) = 1$). In the second plot, we compute the optimal LQ controller $\widehat{\boldsymbol{K}}_T$ using the estimated system $\widehat{\boldsymbol{\theta}}_T = [\widehat{\boldsymbol{A}}_T\ \widehat{\boldsymbol{B}}_T]$ and show the frequency that the synthesized controller is stable, i.e., $\rho(\boldsymbol{A}^\star + \boldsymbol{B}^\star\widehat{\boldsymbol{K}}_T) < 1$.

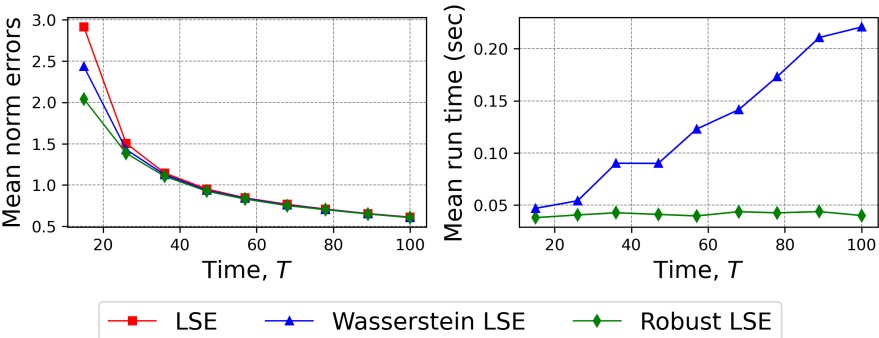

Figure 6: Comparison between our robust LSE model (green) and the Wasserstein model (blue), with both models fine-tuned using cross-validation. The Robust model demonstrates superior performance in terms of mean norm errors. As shown in the right subplot, the run time for our model remains constant regardless of sample size while the run time for the Wasserstein model increases with sample size.

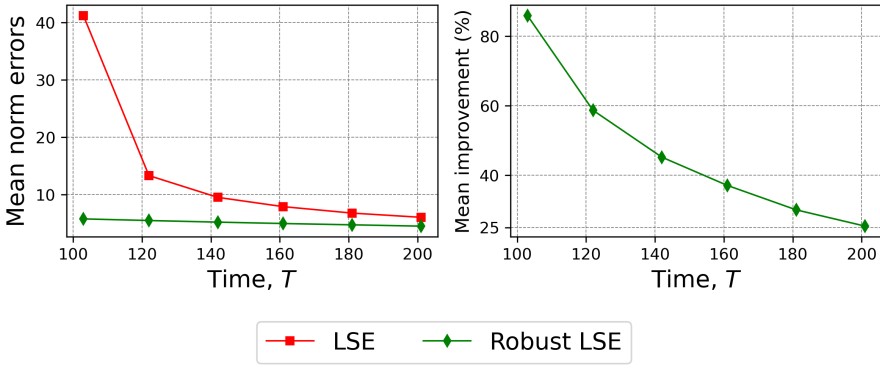

Figure 7: Performance comparison in a large-scale system: we compare our proposed method (green) with the standard LSE (red) in a large-scale linear system where $\boldsymbol{\theta}^\star \in \mathbb{R}^{50 \times 100}$. Robust LSE shows superior performance, similar to other experiments in this paper.

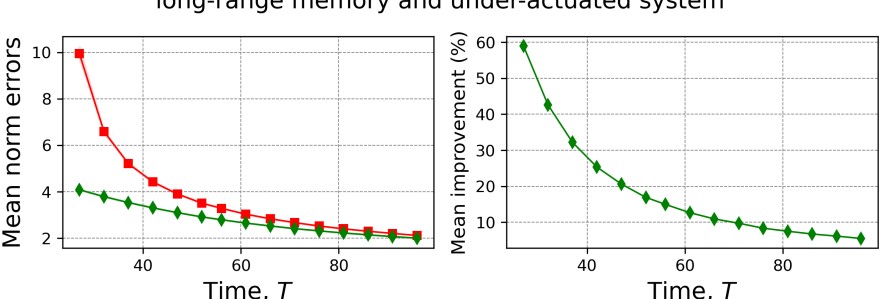

Figure 8: Performance comparison with long-range memory and under-actuated system: the system has $\rho(\boldsymbol{A}^\star) = 0.995$ and $\boldsymbol{A}^\star \in \mathbb{R}^{20 \times 20}$ and $\boldsymbol{B}^\star \in \mathbb{R}^{20 \times 2}$. The results show the superior performance of ours (green) over LSE (red).

# G    MORE ONLINE LQ CONTROL RESULTS

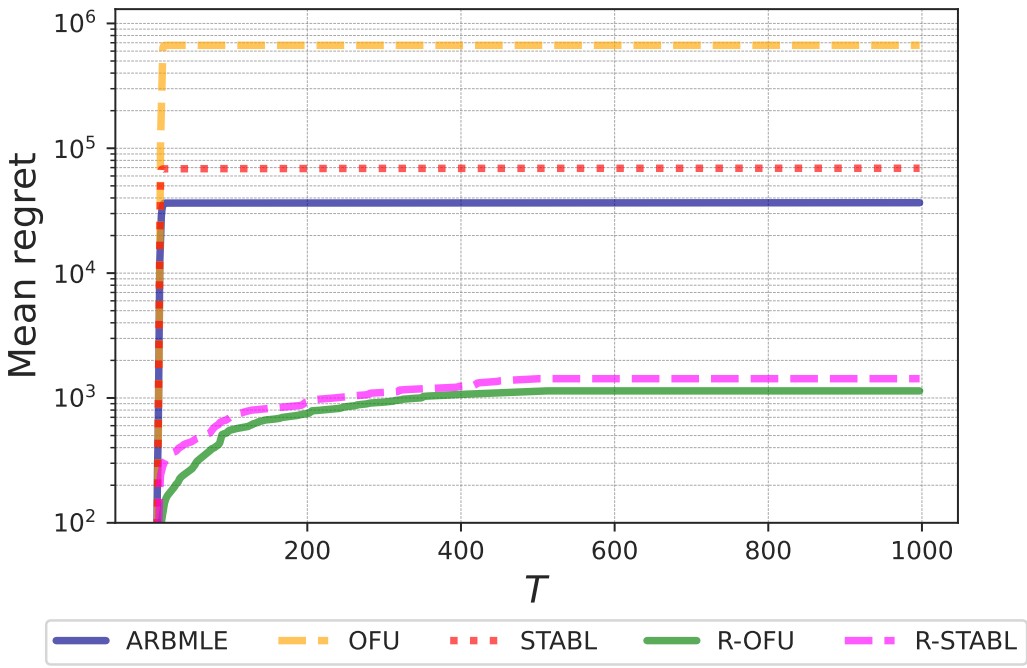

Figure 9: Mean regret over 500 replications: **ii)** marginally unstable Laplacian system:

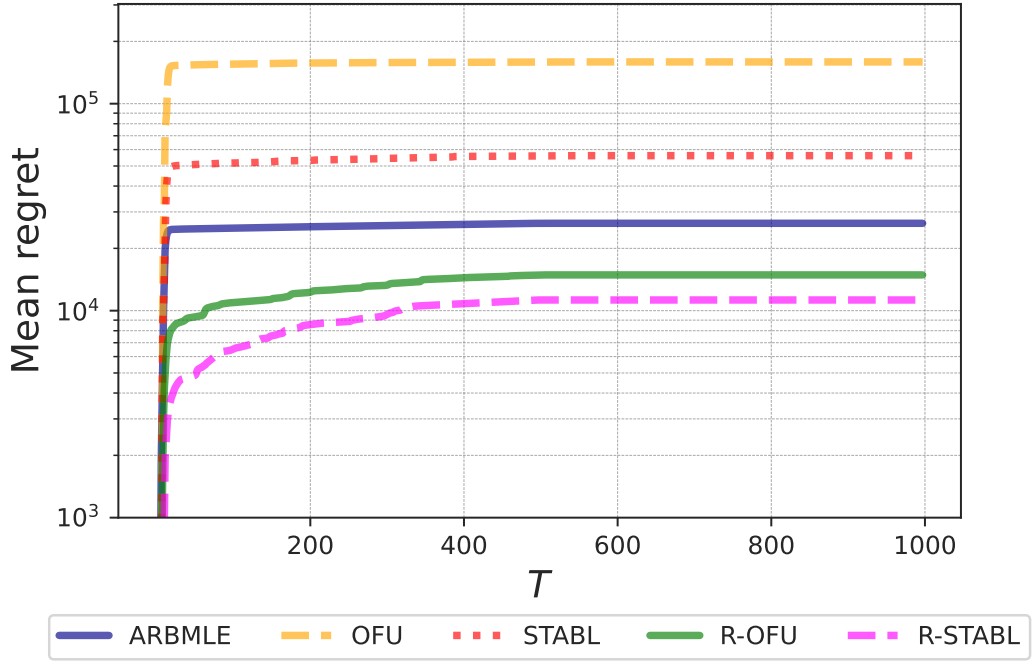

Figure 10: Mean regret over 500 replications: **iii)** UAV in a 2D plane

## H  CHOICE OF THE SCHATTEN NORM PARAMETER $q$

Since the regularization term in (7) is defined using the Schatten norm, the parameter $q$—or equivalently $p$ in (6)—can be chosen based on the distribution of the eigenvalues of the underlying system. To illustrate this relationship, we conducted stylized experiments, as shown in Figure 11.

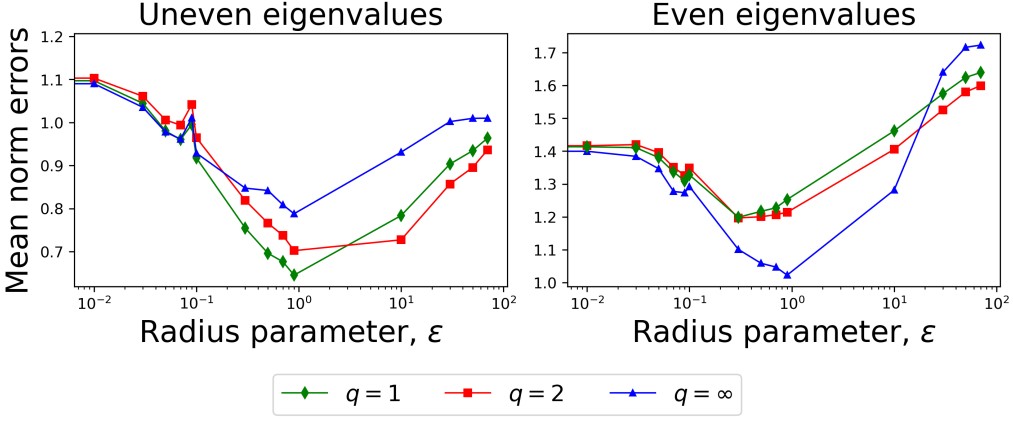

Figure 11: Choosing the optimal Schatten norm parameter $q$ in (7): We generate two $3 \times 3$ linear systems for the experiment. The system in the left subplot has uneven eigenvalues ($\lambda_1 = 0.9$ and $\lambda_2 = \lambda_3 = 0.1$), while the system in the right subplot has even eigenvalues ($\lambda_1 = \lambda_2 = \lambda_3 = 0.9$). We conduct 300 simulations across different choices of the Schatten norm parameter $q \in \{1, 2, \infty\}$ and the radius parameter $\epsilon$ (on the x-axis), plotting the mean norm error (on the y-axis) to identify the optimal radius for each value of $q$. As shown, $q = 1$ performs the best for the system with uneven eigenvalues, while $q = \infty$ performs the best for the system with even eigenvalues.

Furthermore, we validated our findings through adaptive control tasks, as demonstrated in Figure 12.

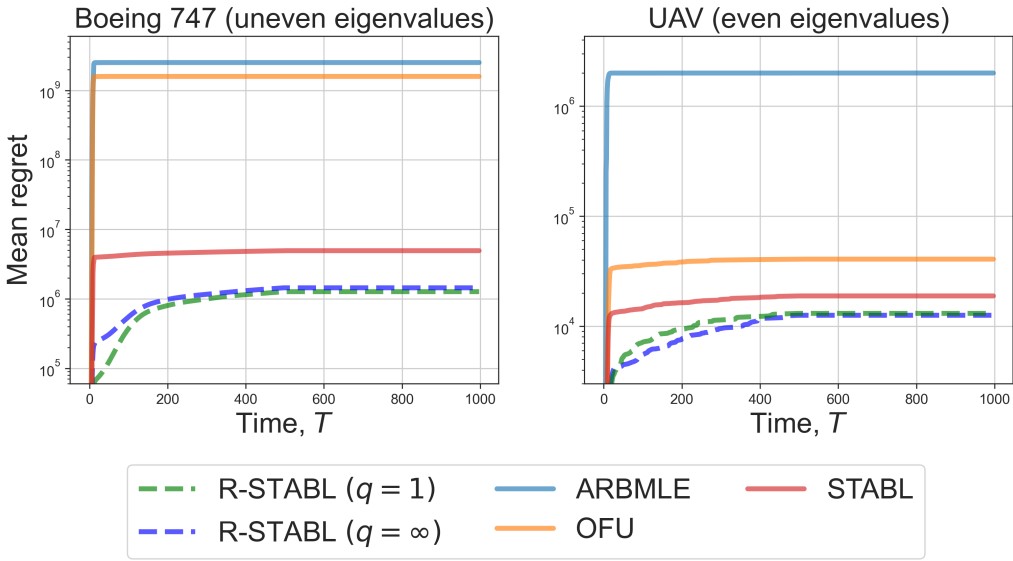

Figure 12: Adaptive control tasks with different choices of $q$: **i)** Boeing 747 represents a system with uneven eigenvalues, whereas **iii)** UAV corresponds to a system with even eigenvalues. We conducted 500 simulations to evaluate adaptive control tasks using different values of $q$ in Robust STABL (R-STABL). The results align with our interpretation of the choice of $q$ as described in Figure 11.

