# OpenReview forum: "Robust System Identification: Finite-sample Guarantees and Connection to Regularization"
_ICLR.cc/2025/Conference — ICLR 2025 Poster_

### Official Review · Reviewer_pX9E · 2024-10-29

**Soundness:** 3
**Presentation:** 2
**Contribution:** 2
**Rating:** 6
**Confidence:** 3

**Summary:**

This paper introduces a robust least squares estimation (LSE) framework that combines robust optimization with the LSE approach to learn nonlinear dynamical systems from a single sample trajectory, demonstrating its equivalence to regularizing LSE with a general Schatten
p-norm. For the linear case, the authors provide non-asymptotic performance guarantees of the framework under certain conditions and show that the error rate is invariant to system dimensionality.

**Strengths:**

The authors propose a robust LSE framework to learn nonlinear dynamical systems and demonstrate its equivalence to the LSE problem regularized by the Schatten p-norm. They also provide theoretical performance guarantees for the framework applied to linear dynamical systems under certain conditions。

**Weaknesses:**

While the paper considers nonlinear dynamical systems, the theoretical results on convergence are limited to linear cases. Additionally, the paper lacks comparisons with other classical methods, such as LSE regularization with special cases of the Schatten
p-norms mentioned in line 54 and the Wasserstein robust optimization discussed in line 59.

**Questions:**

1. In line 162, the authors state that they can express the true LSE problem by substituting $\hat\Omega_T$ in equation 3 with its expectation. Is there any literature to support this? What do you mean by "the true LSE problem"? In my opinion, it is more reasonable to obtain the true LSE problem by substituting $x$ and $\phi$ in equation 2 with their expectation.
2. How do the authors prove that the optimal solution to equation 5 is $\theta^*$?
3. The proposed robust LSE framework is closely related to the Schatten p-norm. Hence, what is the value of p in your experiments? In particular, it is necessary to show the performance of the robust LSE across different norms.
4. In Sections 5.1 and 5.2, the authors only compare the proposed method with the LSE. Comparisons with other classical methods are highly recommended, such as LSE regularization with special cases of the Schatten p-norms mentioned in line 54 and the Wasserstein robust optimization discussed in line 59.
5. It seems that the definition of $\Sigma_\omega$  in equation 9 is missing.

---

> ### Author Response · Authors · 2024-11-15
>
> ${\color{black}\text{1. Theoretical results}}$
> - We agree with the reviewer that our analysis was limited to the linear case. As we acknowledged in the paper, despite its apparent simplicity, the linear system remains challenging to analyze. Most research in statistical learning for system identification has focused on linear systems, as this setting allows for more tractable theoretical analysis.
>
> - However, we’d like to remark that a nonlinear setting, in fact, spans a very broad regime. Several studies have provided non-asymptotic guarantees for subclasses of nonlinear systems that exhibit near-linear behavior. While we appreciate and could have adopted these approaches, their analyses fundamentally rely on techniques similar to those used in linear settings, leveraging near-linear behaviors. Our priority in this paper, however, was to convey the main insight that “regularization or robustification improves empirical performance with nearly no additional theoretical cost (i.e., nearly the same theoretical error)”, encouraging practitioners to adopt our approach over the standard LSE.
>
> ${\color{black}\text{2. Comparison with other Schatten norms AND the Wasserstein robust models}}$
>
> We thank the reviewer for the excellent question.
>
> - As mentioned briefly in the proof of Proposition 1, the choice of $p$ in the Schatten $p$-norm—or equivalently, $q$ in our SDP formulation (7)—has only a minimal impact on the theoretical guarantee, due to the "equivalence of norms."
>
> - However, in practice, the Schatten norm parameter $q$ in our SDP formulation (7) can be selected based on how the eigenvalues of the underlying system are distributed. To illustrate this relationship more clearly, we conducted stylized experiments: ${\color{maroon}\text{for the detail of the experiment, please check Fig. 1 in } \textbf{Supplementary Material}} \text{ or Appendix A.8 in the updated manuscript.}$
>
> - We conducted adaptive control tasks with different choices of $q$: ${\color{maroon}\text{for the detail of the experiment, please check Fig. 2 in } \textbf{Supplementary Material}} \text{ or Appendix A.8 in the updated manuscript.}$
>
> - We further conducted the following experiment where we compared ours with the SOTA Wasserstein model: ${\color{maroon}\text{for the detail of the experiment please check Fig. 3 in } \textbf{Supplementary Material}} \text{ or Appendix A.6 in the updated manuscript}$.
>
> ${\color{black}\text{3. The optimal solution to the true problem}}$
>
> - We believe that the validity of Eq. (5) is a straightforward extension of the consistency of OLS, which is well-documented in many textbooks (e.g., [Shao, 2008]).
> - The following confirms that Eq. (5) holds for $T=1$. For the general case where $T>1$, the same idea applies with only more cumbersome recursion. So we omit that in the rebuttal; a similar derivation for a linear case can be found in [Dean et al., 2020].
>
> First of all, the minimization problem in Eq. (5) is equivalent to
> $$
> \hspace{7cm}
> \min_\theta f(\theta):=\frac{1}{T} \sum_{t=0}^{T-1} \mathbb{E}\left[ \Vert {x}\_{t+1}-\theta \phi\left({z}\_t\right)\Vert_2^2 \right] .
> $$
> which we call the true LSE problem. Recall that our linear system is
> $$
> \hspace{7cm}
> x_{t+1}=\theta^{\star} \phi\left(z_t\right)+w_t .
> $$
> For the case of $T=1$, the objective function is $f(\theta)=\mathbb{E}[\Vert {x}\_1+\theta \phi({z}\_0)\Vert_2^2]$. Then, we have
>
> $
> \hspace{3cm}
> \theta^{\prime} \in \underset{\theta}{\operatorname{argmin}} f(\theta)$ and
> $\frac{\partial}{\partial \theta} f(\theta)=0 \Rightarrow 2 \theta^{\prime}\mathbb{E}[
> \phi\left({z}\_0\right) \phi\left({z}\_0\right)^{\top}
> ] = 2 \mathbb{E}[
> {x}\_1 \phi\left({z}\_0\right)^{\top}
> ].
> $
>
> $\theta^{\prime}=
>  \mathbb{E}[
> {x}\_1 \phi\left({z}\_0\right)^{\top}
> ]
> \left(
> \mathbb{E}[
> \phi\left({z}\_0\right) \phi\left({z}\_0\right)^{\top}]
> \right)\^{-1}$.
> Plugging $x_1=\theta^{\star} \phi\left(z_0\right)+w_0$, we have
>
> $\begin{aligned} \theta^{\prime} & = \theta^{\star} +
> \mathbb{E}\left[
> w_0 \phi\left({z}\_0\right)^{\top}\right] =\theta^{\star}\end{aligned}$
>
> ${\color{black}\text{4. Definition of $\Sigma_w$}}$
>
> - Thanks for your careful review! $\Sigma_w$ was meant to be the covariance of the noise term. We will revise the paper accordingly.
>
>
>
> ${\color{blue}\text{References}}$
> - Shao, Jun. Mathematical statistics. Springer Science & Business Media, 2008.
>
> - Dean, Sarah, et al. "On the sample complexity of the linear quadratic regulator." Foundations of Computational Mathematics 20.4 (2020): 633-679.

---

> > ### Author Response · Authors · 2024-11-25
> >
> > Dear Reviewer pX9E
> >
> > Thank you for your careful comments!
> >
> > Please let us know whether our response addresses your concerns. If so, we kindly ask you to consider raising the rating of our work.
> >
> > We appreciate your time! We are looking forward to discussing any additional concerns you may have.
> >
> > Best wishes,
> >
> > Authors

---

> > > ### Comment · Reviewer_pX9E · 2024-12-03
> > >
> > > Thank you for answering my questions. I increase my score accordingly.

---

### Official Review · Reviewer_X6xt · 2024-11-02

**Soundness:** 3
**Presentation:** 3
**Contribution:** 2
**Rating:** 6
**Confidence:** 3

**Summary:**

1. An equivalent SDP formulation for robust LSE with Schatten p-norm regularization
2. finite-sample guarantees for system identification with single trajectory

**Strengths:**

1. very nice and solid theory
2. provide theoretical guarantees for a hard system identification problem

**Weaknesses:**

1. Is single trajectory a common case in real applications, although it might be an interesting theoretic topic? Could you please provide more motivate examples? Online control of linear system might be one, however the existing works most focus on learning a controller directly rather than following the conventional pipeline (i.e. system identification and then model-based control). What is the advantage of conventional pipeline in the low data region?
2. The authors claim that unlike the SOTA Wasserstein robust optimization, their approach can avoid the curse of dimensionality. However, no comparison study w.r.t. the SOTA method is provided in the paper. And the examples are of very low dimension. The largest linear model is in Sec. 5.2 with $\theta \in \mathbb{R}^{5\times 10}$.
3. Very strong theory but very weak experiments. Most of them are either synthetic or low-dimension datasets.
4. The results in Sec.4 rely on strong assumptions. For example, if the application is online control or wind prediction, then A2 and A6 do not hold.

**Questions:**

1. Due to the setup (perfect state measurement and known model $\phi$), Thm. 1 holds with no significant difference for linear and nonlinear $\phi$. What if the case where only $y_t=x_t+v_t$ is available or $\phi$ has some uncertainty? For simplicity, we can assume that $\phi$ is linear.

---

> ### Author Response · Authors · 2024-11-18
>
> ${\color{black}\text{1. Motivating examples of the single trajectory setup}}$
>
> We thank the reviewer for the excellent question.
>
> - We believe that the single-trajectory setting more accurately reflects the realistic data collection process: in dynamic systems, data collection naturally results in non-iid data since each new observation depends on the current state.
>
> - In this context, the iid data assumption is more restrictive, namely, the iid assumption is that we have $N$ iid trajectories, $\\{x_t^i\\}\_{t=0}^T$ for $i = 1, \ldots, N$. As noted in [Dean et al., 2020], the iid assumption implies that we somehow have the ability to reset the “unknown” system to a state independent of past observations (e.g., $x_0 = 0$).
>
> -  While resetting a system may be straightforward in certain robotics experiments—where an agent can easily be returned to an initial state such as a specific position or angle—this is not always feasible. For example, in safety critical missions such as drones deployed in space or underwater, data-collection is very limited.
>
> - This is also related to the advantage of conventional pipelines. A model-free approach can provide very flexible policies that a model-based approach may not come up with. However, the model-based approach is more data-efficient, hence, more suitable under a low data regime.
>
> ${\color{black}\text{2. Comparison with the SOTA method AND in a large scale system}}$
>
> We really appreciate the constructive feedback from the reviewer.
>
> -  We’d like to point out that wind prediction is a real-world problem that has been considered by various works.
>
> - Following the suggestion we conducted the following experiments with a large scale system with dimensions being $50\times100$: ${\color{maroon}\text{please check Fig. 5 in } \textbf{Supplementary Material}} \text{ or Appendix A.7 in the updated manuscript}$. The results not only align but also corroborate our previous experiments.
>
> - We further conducted the following experiment where we implemented the SOTA Wasserstein model: ${\color{maroon}\text{please check Fig. 3 in } \textbf{Supplementary Material}} \text{ or Appendix A.7 in the updated manuscript}$.
>
> $\quad$ - Our method achieves better performance in mean norm error. We also demonstrate the computational advantage of our model: the run time for our SDP formulation remains constant with sample size, whereas the run time for the Wasserstein model scales with sample size.
>
> ${\color{black}\text{3. Assumptions in Sec. 4}}$
> - First, we’d like to point out that A2 is a standard assumption in this stream of research as described in the review paper [Tsiamis et al., 2023].
>
> - That said, we agree that A6 may be too strong. In fact, A6 is a simplifying assumption intended to improve the readability of our paper and avoid overly cumbersome notation in our theoretical derivations. A more realistic assumption in this line of research is the following:
> we have access to some policy $K_0$ that stabilizes the unknown system. As stated in [Dean et al., 2020], in many cases a stabilizing policy can be found efficiently. With this assumption replacing A6, we can inject exploration noise into the stabilizing policy, i.e., $u_t = K_0 x_t + \eta_t$, where $\eta_t$ is sub-Gaussian noise, and derive similar guarantees to Proposition 1.
>
> ${\color{black}\text{4. Extension to unobservable state setting}}$
>
> We thank the reviewer for the interesting question.
>
> - As briefly mentioned in Conclusion, extending the proposed approach to unobservable systems is a direction for future research.
>
> - In [Oymak and Necmiye, 2019], the authors consider an unobservable linear time-invariant system:
> $$
> \hspace{7cm}
> \begin{aligned}
>  \boldsymbol{x}_{t+1}=\boldsymbol{A} \boldsymbol{x}_t+\boldsymbol{B} \boldsymbol{u}_t+\boldsymbol{w}_t  \\\\
> \boldsymbol{y}_t=\boldsymbol{C} \boldsymbol{x}_t+\boldsymbol{D} \boldsymbol{u}_t+\boldsymbol{z}_t.
> \end{aligned}
> $$
> They show that the standard LSE can be used to recover the Markov parameters of the linear system and reconstruct $A, B, C, D$ from the Hankel matrix via the Ho-Kalman algorithm. Using this idea, we can derive a similar SDP formulation as in Eq (7) for such a system.
>
> - While the extension to the unobservable setup is direct from an algorithmic development standpoint, we are unsure whether a new approach is required to derive theoretical results. It would be interesting to explore if we can establish theoretical guarantees for our approach in this setup.
>
>
>
> ${\color{blue}\text{References}}$
>  - Dean, Sarah, et al. "On the sample complexity of the linear quadratic regulator." Foundations of Computational Mathematics 20.4 (2020): 633-679.
>
> - Tsiamis, Anastasios, et al. "Statistical learning theory for control: A finite-sample perspective." IEEE Control Systems Magazine 43.6 (2023): 67-97.
>
> - Oymak, Samet, and Necmiye Ozay. "Non-asymptotic identification of LTI systems from a single trajectory." 2019 American control conference (ACC). IEEE, 2019.

---

> > ### Comment · Reviewer_X6xt · 2024-11-22
> > **Strong theory but weak experiments**
> >
> > Thanks for your responses. I am still not fully convinced by part of the responses and the new experimental results.
> >
> > 1. Single trajectory setup
> > - I don't understand why the mutli-trajectory cases should be iid. Even through, I still don't agree with the argument "in dynamic systems, data collection naturally results in non-iid data". If you mean that it is non-iid between different $t$, then most system identification algorithm does not have such assumption. If you mean that it is non-iid between different $i$, then I don't think it is difficult to satisfy for dynamical systems. Let's say, if I turn off the power of a robot after collecting $\{x_t^1\}$, then I redo some configuration setup,  turn it on, and collect a new trajectory $\{x_t^2\}$. Why it is not iid between different trajectories? Moreover, if the system has to be continuously operating, based on your assumption $A$ is stable, it means that initial condition is forgot exponentially. So iid trajectories can be obtained by stopping data collection for a while and then restarting it.
> > - Could you provide some references to support your claim that model-based method (identification and then control design) is more sample-efficient than directly learning a policy, providing that the system is already stable?
> >
> > 2. Experiment comparison
> > - I understand that your increase the system size by 10 times. But it is still very small (state dimension is 50) and the system is fully-actuated ($A,B\in \mathbb{R}^{50\times 50}$).
> > - The Wasserstein LSE is based on Esfahani & Kuhn (2018). I don't doubt that their theoretic results are SOTA. But the computational side is not necessary, providing the recent progresses in computational optimal transport.
> > - The Wasserstein LSE is a large linear programming, where the number of constraints depends on the sample size. It's obvious that the computation cost will increase with the sample size. But the computation time can be significantly reduced if one applies the mini-batch approach and perform optimization on GPU. So far, SDP is not GPU friendly.
> > - There are lots of GPU-friendly distributionally robust optimization (DRO) methods published in those ML conferences since 2018. However,  the comparison is done for Wasserstein LSE which is a CPU-based solver without implementing mini-batch schemes.
> > - The SDP approach scales very well with sample size. But it scales poorly with the dimension (i.e., $\mathrm{dim}(P)$ matters if $P\succeq 0$). However, the Wasserstein approach may scale well for this.
> >
> > **Summary:** I am satisfied with the responses for Points 3 \& 4. I really like the theory part but feel that the experiments are a little bit sloppy. I will consider to raise my score if
> >
> > 1. Provide a comparison study on the dimension size. Fix the sample size while increasing the dimension of $(x,u)$, e.g., (50, 20), (500, 200), (5000, 2000). Compare the computational cost.
> > 2. Explore the performance of the proposed approach on the long-range memory and under-actuated case. That is, long-range memory means $\eta \leq \rho(A)<1$ with $\eta$ is very close to 1, e.g. $\eta=0.995$. ``Under-actuated" means $\mathrm{dim}(u)<0.1\mathrm{dim}(x)$.
> > 3. Provide a limitation discussion paragraph later, if the authors do not have time to do comparison with more recent DRO methods.

---

> > > ### Comment · Reviewer_Wdgh · 2024-11-22
> > >
> > > If I may interject to provide a few on-hand references for Reviewer X6xt:
> > > - Though running a typical sysID algorithm does not require foreknowledge of the dependency structure, analyzing the non-asymptotic risk of such algorithms (e.g. LSE in Simchowitz et al, Jedra and Proutiere) is typically more involved. Notably, Dean et al. assume access to IID *trajectories*, and notably their estimator only uses the last iterate of each trajectory, and thus their risk bounds only scale with the number of trajectories $N$ and not with the length of each trajectory $T$, whereas single-trajectory-type analysis would yield joint scaling in both ($N \to NT$) in that setting. Therefore, even with the ability to reset, single-trajectory analysis is tighter in that you don't need to throw away your dependent data. Whether the necessary assumptions to establish this kind of analysis is satisfied in real-life robotic settings is a different story, though.
> > > - A popular paper formalizing the sample-efficiency of a "model-based" plug-in approach vs "model-free" policy optimization approach for Linear-Quadratic settings is Tu and Recht; see e.g. page 9 in https://arxiv.org/abs/1812.03565. As far as I know, these gaps remain even if the nominal system is stable $\rho(A) < 1$.
> > >
> > > Simchowitz et al., Learning Without Mixing: Towards A Sharp Analysis of Linear System Identification
> > >
> > > Jedra and Proutiere., Finite-time Identification of Stable Linear Systems Optimality of the Least-Squares Estimator
> > >
> > > Dean et al., On the sample complexity of the linear quadratic regulator
> > >
> > > Tu and Recht, The Gap Between Model-Based and Model-Free Methods on the Linear Quadratic Regulator: An Asymptotic Viewpoint

---

> > > > ### Comment · Reviewer_X6xt · 2024-11-22
> > > >
> > > > Thank you, Reviewer Wdgh, for providing further explanations and references. Then, I do not have any further theoretic questions. I will increase my score if the authors can provide some further experimental results.

---

> ### Author Response · Authors · 2024-11-22
>
> We thank X6xt for the valuable feedback and Wdgh for providing further explanations and references. The rebuttal process has been very helpful for us to identify areas we may have overlooked. In particular, we are less familiar with the literature on improving the computational aspects of DRO using GPUs, so it’s great to learn about such efforts. Our comparison was performed assuming that the system operator has access only to standard off-the-shelf solvers.
>
> - First, we’d like to clarify that the Wasserstein model used in our experiments is not an LP,  but an SDP, since the objective function is squared errors. As the reviewer mentioned, we could implement the LP version of the Wasserstein model by changing the objective function to absolute errors. However, our experiments were intended to compare the Wasserstein "LSE" model with our robust LSE model. In this case, the Wasserstein model shares the same complexity as our formulation, even when the sample size is fixed. Unfortunately, we are not aware of Wasserstein LSE models that have LP or SOCP formulations.
>
> - So, we certainly agree with the reviewer’s concern. In our tests with linear systems with a state dimension being 50, solving the SDP formulation using MOSEK (solver) took an average of 40 seconds. Therefore, for such large systems, we agree that an SDP formulation is not a practical choice to begin with.
>
> - That said, as mentioned in our paper, the SDP formulation in Equation (7) can be reduced to an unconstrained quadratic program (QP) by setting $q = 1$. While this approach loses the flexibility of selecting the Schatten norm parameter, it certainly offers a reasonable tradeoff given the computational concerns. On the other hand, the Wasserstein model does not admit such regularization interpretation for a least squares loss function so its formulation remains not scalable.
> We will discuss the limitations of our approach and SDP formulations in the paper.

---

> > ### Author Response · Authors · 2024-11-22
> >
> > Following the suggestion from the reviewer X6xt, we've conducted the experiment where we consider the long-range memory and under-actuated system where $A^\star\in\mathbb{R}^{20\times20}, B^\star\in\mathbb{R}^{20\times2},$ and $\rho(A^\star)=0.995$: ${\color{maroon}\text{please check Fig. 6 in } \textbf{Supplementary Material}}.$ The results show the superior performance of our method.

---

> > ### Comment · Reviewer_X6xt · 2024-11-22
> >
> > Thank you for addressing my concerns. I am happy to increase my score.

---

### Official Review · Reviewer_JZsQ · 2024-11-03

**Soundness:** 3
**Presentation:** 3
**Contribution:** 3
**Rating:** 6
**Confidence:** 4

**Summary:**

The authors study the problem of identifying non-linear system dynamics from a single trajectory. They introduces a robust framework for system identification and reveal the connection to regularized empirical risk minimization. They provide non-asymptotic guarantees for linear systems and demonstrate an improved error rate of $\tilde{\mathcal{O}}(1/\sqrt{T})$, which the authors claim avoids the curse of dimensionality faced by other robust methods such as Wasserstein robust optimization. Empirical results validate the effectiveness of robust LSE in system identification and online control tasks.

**Strengths:**

**Clarity of exposition:** The paper is well-written and identifies the limitations of the standard LSE in system identification under limited data or misspecified models, providing motivation for the proposed robust LSE.

**Theoretical contribution:** The paper presents non-asymptotic error bounds that establish the effectiveness of Schatten $p$-norm regularization, positioning the method as a theoretically grounded alternative to Wasserstein robust optimization.

**Empirical Validation:** Several numerical experiments are conducted, including tasks in wind speed prediction and online control, with substantial improvements over classical LSE.

**Weaknesses:**

**Discussion of the theoretical results:** The authors very briefly discuss their main results and the implications of the regularization term in their bounds. In particular, it is important to note that $T$ still scales with the system dimensions, i.e., $n+m$ in Theorem 2. It is important to make clear the difference between the curse of dimensionality and the unavailable dependence of the system dimension in the burn-in.

**Questions:**

1) The authors mention the use of different Schatten norms but does not provide insight into the practical or theoretical consequences of varying $p$. Could the authors expand more on this?

2) Could the authors provide more intuitions on the key factors for avoiding the course of dimensionality in their results? The way the error bounds is presented in Theorem 2 it is not clear the benefit of the regularization term.

---

> ### Author Response · Authors · 2024-11-18
>
> ${\color{black}\text{1. The practical or theoretical consequences of varying Schatten norm parameter $p$ }}$
>
> We sincerely thank the reviewer for the constructive feedback.
>
> - As mentioned briefly in the proof of Proposition 1, the choice of $p$ in the Schatten $p$-norm—or equivalently, $q$ in our SDP formulation (7)—has only a minimal impact on the theoretical guarantee, due to the "equivalence of norms."
>
> - However, in practice, the Schatten norm parameter $q$ in our SDP formulation (7) can be selected based on how the eigenvalues of the underlying system are distributed. To illustrate this relationship more clearly, we conducted stylized experiments: ${\color{maroon}\text{please check Fig. 1 in } \textbf{Supplementary Material}} \text{ or Appendix A.7 in the updated manuscript.}$
>
> $\quad$ - We generate two $3 \times 3$ linear systems for the experiment. The system in the left subplot has uneven eigenvalues ($\lambda_1 = 0.9$ and $\lambda_2 = \lambda_3 = 0.1$), while the system in the right subplot has even eigenvalues ($\lambda_1 = \lambda_2 = \lambda_3 = 0.9$). We conduct 300 simulations across different choices of the Schatten norm parameter $q\in\\{1,2,\infty\\}$ and the radius parameter $\epsilon$ (on the x-axis), plotting the mean norm error (on the y-axis) to identify the optimal radius for each value of $q$. As shown, $q = 1$ performs the best for the system with uneven eigenvalues, while $q = \infty$ performs the best for the system with even eigenvalues.
>
> - We further conducted adaptive control tasks with different choices of $q$: ${\color{maroon}\text{please check Fig. 2 in } \textbf{Supplementary Material}} \text{ or Appendix A.7 in the updated manuscript.}$
>
> $\quad$ - For example, $\textbf{i)}$ Boeing 747 represents a system with uneven eigenvalues, whereas $\textbf{iii)}$ UAV corresponds to a system with even eigenvalues. We conducted 500 simulations to evaluate adaptive control tasks using different values of $q$ in Robust STABL (R-STABL). The results align with our interpretation in Fig. 1.
>
> ${\color{black}\text{3. Avoiding the curse of dimensionality}}$
>
> We thank the reviewer for the clarifying question.
>
> -  We may mislead the reviewer in using the term “curse-of-dimensionality” when comparing our result with the Wasserstein robust model.  As mentioned, the Wasserstein model encounters the curse-of-dimensionality because its error rate is $\mathcal{O}(1/T^{\frac{2}{n}})$ where $T$ is the number of samples and $n$ is the dimension of state space. We used the term “the curse-of-dimensionality” to describe this exponential dependence of $T$ (i.e., # of sample) on $n$, which is not present in the guarantee provided by our robust model. Addressing this exponential dependence is an open research topic that the robust optimization community tries to address. Although a different approach was proposed by [Gao, 2023] under the assumption of iid data and Lipschitz continuity with respect to $\theta$, this approach is not applicable to our system identification problem.
>
>
> ${\color{black}\text{4.  The difference between the curse of dimensionality and the unavailable dependence of the system dimension in the burn-in}}$
>
> We appreciate the reviewer for the careful feedback.
>
> - As another reviewer suggested, $T$ scales with other problem-dependent parameters, most notably the dimension of the system, i.e., $n$ and $m$. We acknowledge that it would have been more informative to present the explicit upper bound for the radius parameter $\epsilon(\delta)$ in Eq. (11). We have simplified the explicit bound as shown below:
>
> $$
> \epsilon(\delta) = c_1\left(\frac{\sqrt{\log \left(\frac{2}{\delta}\right)+c_2 n}}{T^{\frac{3}{4}}}+\frac{\left(\log \left(\frac{2}{\delta}\right)+c_2 n\right)}{T}\right)+c_3 \frac{1}{\sqrt{T}} \cdot \sqrt{\log (T)+\log \left(\frac{5^n}{\delta}\right)}
> +c_4\left(\sqrt{\frac{n+m}{T}}+\frac{n+m}{T}+
> \left.
> \sqrt{\frac{\log \left(\frac{c_5}{\delta}\right)}{T}}
> +
> \frac{\log \left(\frac{c_5}{\delta}\right)}{T}\right)\right.
> $$
> Here, the constants $c_1,\\;c_2,\\;c_3,\\;c_4,\\;c_5$ are universal constants.
>
>
>
> ${\color{blue}\text{References}}$
> - Gao, Rui. "Finite-sample guarantees for Wasserstein distributionally robust optimization: Breaking the curse of dimensionality." Operations Research 71.6 (2023): 2291-2306.

---

> > ### Comment · Reviewer_JZsQ · 2024-11-21
> > **Official comment by Reviewer JZsQ**
> >
> > Thank you to the authors for their thorough response. I'm happy to increase my score.

---

### Official Review · Reviewer_Wdgh · 2024-11-04

**Soundness:** 3
**Presentation:** 3
**Contribution:** 3
**Rating:** 8
**Confidence:** 4

**Summary:**

In this paper, the authors propose a novel estimator for discrete-time, single-trajectory system-identification via Schatten-norm regularization. The authors claim that the proposed estimator combines the desiderata of rate-optimal guarantees of the standard least-squares estimator and robustness to misspecification a la Wasserstein robustness. The authors support their claims on some numerical experiments, demonstrating that in the presence of misspecification (e.g. from nonlinear dynamics), the proposed estimator has better convergence properties than the standard LSE.

**Strengths:**

The paper is in general well-written and straightforward to understand. The method derivation is quite clean, demonstrating that min-max robust LSE can be turned into an semi-definite program, which can then be reduced further to an unconstrained Schatten-norm-regularized least-squares problem. Something of note is that the proposed SDP only interacts with the trajectory through the empirical Gram matrix, and thus does not need to scale with the length of the trajectory. The theoretical guarantees of the estimator also demonstrate the rate-optimality of the procedure, which is otherwise not true for out-of-the-box guarantees of Wasserstein robust estimators. Numerical experiments demonstrate that not only is the estimator performant compared to LSE, it can be slotted into various online control frameworks to great effect.

I think the paper is good enough for acceptance; I have bumped my score up after the author's response.

**Weaknesses:**

Since the proposed method is a regularized version of the least-squares estimator, it inherently has an additional moving part in the regularization parameter. In theory, for the robustness set to cover the population Gramian with high probability, the regularization parameter is determined by the convergence rate of the empirical Gramian to the population one, and thus can only be as good as the (typically conservative) upper bounds. Therefore, the authors propose as a heuristic to determine the best scaling of the regularization parameter via cross-validation, and demonstrate that this tuned method outperforms LSE by a large relative amount. However, this is somewhat unenlightening, as this is comparing a fine-tuned algorithm to *the* base estimation algorithm (which is technically subsumed by the proposed class of estimators). It would be more informative if the comparison were to a different class of (fine-tuned) robust estimators.

The paper also assumes strict stability. While this is likely for technical reasons in the proof, as the authors follow Jedra and Proutiere's (2020) Hanson-Wright approach, something that is left wanting in this paper is whether proposed estimator also performs better in the face of (near) marginal stability, since the authors point out in Section 5.2 that the estimator actually performs relatively better in the very-stable regime. In the same vein, it would be more enlightening to see stabilization rate of synthesized controllers from the estimated matrices $\hat A, \hat B$ as a function of data, e.g. via certainty-equivalent LQR, as this is a more practical notion of estimator robustness.

The paper also contains a few confusing parts. Most notably, the uncertainty level $\epsilon(\delta)$ in (11) is set with minimal details on the parameters therein. Since this quantity shows up in the error bound in Theorem 2, it would be good to at least show the valid upper bound on $\epsilon(\delta)$ derived in the appendix, since the main paper suggests $\epsilon(\delta) = \tilde{\mathcal O}(1/\sqrt{T})$ is the quantity up to universal constants, while in reality it contains some (rather important) system-dependent quantities.

**Minor comments**
- Should provide mathematical definition of subgaussianity in the assumptions.

- Should define what $\|\nabla_\theta \|\mathbf A(\theta^\star)\|_q\|$ is in (12). Also, this seems to differ from the quantity in the appendix proof (42).

Jedra and Proutiere, "Finite-time Identification of Stable Linear Systems Optimality of the Least-Squares Estimator", 2020.

**Questions:**

Please see the above Weaknesses section for areas to clarify.

Also, it would be good to expand on the comparison to the DRO literature. Notably, why is (Gao, 2023) not applicable to even the iid version of this least-squares problem? For example, if we instead pretend to draw each point of the trajectory from its marginal distribution and apply a beta-mixing blocking argument, would (Gao, 2023) be applicable to the independent process?

Rui Gao, "Finite-sample guarantees for wasserstein distributionally robust optimization: Breaking the curse of dimensionality", 2023.

Kuznetsov and Mohri, "Generalization bounds for non-stationary mixing processes", 2016.

---

> ### Author Response · Authors · 2024-11-18
>
> ${\color{black}\text{1. Comparison with the SOTA method}}$
>
> We really appreciate the constructive feedback from the reviewer.
>
> - Indeed, other reviewers suggested comparing our method against the fine-tuned SOTA Wasserstein robust model: ${\color{maroon}\text{please check Fig. 3 in } \textbf{Supplementary Material}} \text{ or Appendix A.7 in the updated manuscript}$.
>
> $\quad$ - Our method achieves better performance in mean norm error. We also demonstrate the computational advantage of our model: the run time for our SDP formulation remains constant with sample size, whereas the run time for the Wasserstein model scales with sample size.
>
> ${\color{black}\text{2. Comparison with a marginally stable system}}$
>
> Thanks for the excellent suggestion.
> - Following the reviewer's suggestion, we further conducted the experiment with a marginally stable system and simulated the stability: ${\color{maroon}\text{please check Fig. 4 in } \textbf{Supplementary Material}} \text{ or Appendix A.7 in the updated manuscript.}$
>
> $\quad$ - As the reviewer suggested, the results seem to illustrate that the stabilization rate and system errors are related.
>
> ${\color{black}\text{3. Comparison to the DRO literature}}$
> - The analysis in [Gao, 2023] requires Lipschitz continuity of the loss function and boundedness of the feasible set. Neither is satisfied in our setting, and it's unclear how the proofs can be extended to our setting.
>
>
> ${\color{black}\text{4. The valid upper bound on $\epsilon(\delta)$}}$
>
> We thank the reviewer for the careful feedback
>
> - We acknowledge that it would have been more informative to present the explicit upper bound for the radius parameter $\epsilon(\delta)$ in Eq. (11). We have simplified the explicit bound as shown below:
>
> $$
> \epsilon(\delta) = c_1\left(\frac{\sqrt{\log \left(\frac{2}{\delta}\right)+c_2 n}}{T^{\frac{3}{4}}}+\frac{\left(\log \left(\frac{2}{\delta}\right)+c_2 n\right)}{T}\right)+c_3 \frac{1}{\sqrt{T}} \cdot \sqrt{\log (T)+\log \left(\frac{5^n}{\delta}\right)}
> +c_4\left(\sqrt{\frac{n+m}{T}}+\frac{n+m}{T}+
> \left.
> \sqrt{\frac{\log \left(\frac{c_5}{\delta}\right)}{T}}
> +
> \frac{\log \left(\frac{c_5}{\delta}\right)}{T}\right)\right.
> $$
> Here, the constants $c_1,\\;c_2,\\;c_3,\\;c_4,\\;c_5$ are universal constants.
>
> ${\color{blue}\text{References}}$
> - Gao, Rui. "Finite-sample guarantees for Wasserstein distributionally robust optimization: Breaking the curse of dimensionality." Operations Research 71.6 (2023): 2291-2306.

---

> ### Comment · Reviewer_Wdgh · 2024-11-21
>
> Thank you for the reply. The authors have addressed my main comments satisfactorily, and I have increased my score. I am particularly satisfied by the % stabilized experiment (Figure 10), which concretely shows the "safety" improvement from using the proposed robust estimator for sysID + certainty equivalent control over naive LSE, even in the face of (nominal) marginal stability.
>
> I do agree with Reviewer q1qq that misspecification is a very important desideratum that is largely left open. However, I'm willing to view this as too large a can of worms for this paper, as it is not obvious to me the best way to quantify "misspecification" in a non-trivial, control-oriented way beyond a naive norm perturbation. Furthermore, the aforementioned stabilization experiment partially assuages this point for me, as one can view *certainty equivalent control* on the identified dynamics as a proxy for performance under misspecification.

---

### Official Review · Reviewer_q1qq · 2024-11-05

**Soundness:** 3
**Presentation:** 3
**Contribution:** 3
**Rating:** 5
**Confidence:** 4

**Summary:**

Building on recent work by authors such as Simchowitz, Jedra and others, this paper
studies system identification for linear systems in the one trajectory setup,
showing that a regularized form of least-squares estimation (LSE)---with the
regularization being a Schatten $p$-norm---has favorable nonasymptotic convergence,
and (empirically) outperforms unregularized LSE on some standard benchmarks.

**Strengths:**

The problem is a significant one, and the theoretical results presented by the authors are nicely executed and nicely presented.

**Weaknesses:**

Clearly it's appropriate to regularize least-squares estimators in settings in
which there isn't much data, and for matrix estimands a standard matrix norm such
as the Schatten $p$-norm is a natural choice.  This would have seemed to be a
reasonable starting place for the paper.  But, instead of just starting with this
(statistical) perspective, the current paper starts with a robust optimization
(minimax) perspective.  In the authors' words, "seeking the best system parameter
against the worst-case realizations of the data."  But the latter motivation is
rather different from the motivation of small data sets, and it's a bit odd to
start with robustness if the focus is small samples.

The authors do show that the minimax perspective yields the Schatten $p$-norm
regularization.  Perhaps this is just a matter of taste, but this sequence of
logic doesn't make much conceptual sense to me.

A robustness perspective would seem to make sense if the issue is one of misspecification,
and indeed the authors state that "it [the LSE] suffers from poor identification
errors when the sample size is small or the model fails to capture the system's true
dynamics."  It's the latter statement that seems well aligned with the robustness
motivation.  But the paper doesn't seem to follow up on this issue, either theoretically
or empirically (e.g., by considering misspecification).

With respect to the theoretical results, they are indeed very nicely executed and presented,
but I'm not sure how much novelty there is.  My sense is that this is a redeployment of tools
from the papers of Simchowitz, Jedra and others; I'd be happy to stand corrected if that's not the case.

**Questions:**

My main question is regarding the extent to which there is novelty in the proof of the main theorem (Theorem 2).

---

> ### Author Response · Authors · 2024-11-18
>
> ${\color{black}\text{1. Model misspecification}}$
>
> We really appreciate the enlightening feedback from the reviewer.
>
> - Poor out-of-sample (i.e., test) performance caused by small data is one of the primary motivations behind robust optimization. Numerous studies in robust optimization demonstrate that robust solutions enhance out-of-sample performance [Bertismas et al., 2018]. That said, we acknowledge that the logical flow might have been clearer for a general audience if we had begun with the regularization formulation. We will update our manuscript accordingly.
>
> - However, we would like to emphasize that the ideas in this paper originated from an attempt to apply robust optimization to system identification, without initially anticipating its equivalence to the Schatten $p$-norm regularization.
> As the reviewer noted, we did not provide theoretical implications of robustness against model misspecification while the wind prediction experiment was intended to empirically demonstrate this robustness: the neural network model is an approximation of the dynamic of wind speed. In response to the feedback, we have considered the following.
>
> Recall that our linear system (1) is
> $$
> \hspace{7cm}
> x_{t+1}=\theta^{\star} \phi\left(z_t\right)+w_t,
> $$
> and the uncertainty set (6) is
> $$
> \hspace{7cm}
> \mathcal{U}^{p,\epsilon}_T = \left\\{ \Omega: \Vert \Omega - \widehat{\Omega}_T \Vert \leq \epsilon
> \right\\}.
> $$
> Instead of (6), consider the following set:
> $$
> \hspace{3cm}
> \begin{aligned}
> \mathcal{W}_T^{p, \epsilon}=\left\\{ \psi(\cdot): \exists \\; \Omega \\; \text{ s.t. } \\; \Vert \Omega - \widehat{\Omega}_T \Vert \leq \epsilon,
> \\;\\; \Omega=\frac{1}{T}\sum^{T-1}\_{t=0} \begin{bmatrix} x\_{t+1} \\\ \psi(z_t)  \end{bmatrix} \begin{bmatrix} x\_{t+1} \\\ \psi(z_t)  \end{bmatrix}^\top,
> \\;\\; \widehat{\Omega}_T=\frac{1}{T}\sum^{T-1}\_{t=0} \begin{bmatrix} x\_{t+1} \\\ \phi(z_t)  \end{bmatrix} \begin{bmatrix} x\_{t+1} \\\ \phi(z_t)  \end{bmatrix}^\top
>  \right\\}
> \end{aligned}
> $$
>
>  As shown above, instead of a set of the matrix $\Omega$, we can express our uncertainty set as a set of nonlinear functions $\psi(\cdot)$. This alternative expression implies that the uncertainty set constrains the functional space of $\psi(\cdot)$. Therefore, we can interpret our Robust LSE problem in the following:
> $$
> \hspace{3cm}
> \min _\theta \max _{\Omega \in \mathcal{U}_T^{p, \epsilon}} \operatorname{tr}(G(\theta) \Omega) =
> \min _\theta \max _{\psi(\cdot) \in \mathcal{W}_T^{p, \epsilon}}  \operatorname{tr}\left(G(\theta) \left(\frac{1}{T}\sum^{T-1}\_{t=0} \begin{bmatrix} x\_{t+1} \\\ \psi(z_t)  \end{bmatrix} \begin{bmatrix} x\_{t+1} \\\ \psi(z_t)  \end{bmatrix}^\top\right) \right).
> $$
>
> In this context, our problem provides robustness against model misspecification in the nonlinear function $\phi(\cdot)$ if $\phi(\cdot)$ is indeed unknown like the wind speed prediction example in our paper.
>
> ${\color{black}\text{2. Novelty}}$
> - As proposed, our primary goal in this paper is to address the poor empirical performance of LSE under data scarcity and/or model misspecification. We appreciate the reviewer’s suggestion that regularization could be a natural choice to address these issues. However, to our knowledge, no prior papers are providing a theoretical non-asymptotic guarantee for regularized LSE in the standard setup. In this regard, we'd like to emphasize that our results are new while we acknowledge that the derivations rely on standard convex optimization theory and statistical learning theory.
>
> - As briefly noted in our paper, most regularization methods rely on heuristic choices for the regularization parameter, often without a solid theoretical foundation. For instance, the parameter is frequently set to a fixed small value or made independent of the number of samples. In contrast, our results suggest that the regularization parameter should be data-dependent to achieve optimal performance.
>
> - Specifically, Theorem 1 and Corollary 1—which establish the equivalence between regularization and robustness—represent a significant contribution, as this equivalence allowed us to adopt a distinct (i.e., robust) framework to derive non-asymptotic guarantees for regularization, thereby advancing theoretical understanding in the system identification community.
>
> - Our results also have implications for the robust optimization community, as our robust model addresses the curse of dimensionality, which remains a challenge for the state-of-the-art Wasserstein model (${\color{maroon}\text{please check Fig. 3 in } \textbf{Supplementary Material}} \text{ or Appendix A.6 in the updated manuscript}$ for the numerical experiment.) In this way, we believe that our theoretical contributions are relevant to both communities by bridging system identification and robust optimization.
>
> ${\color{blue}\text{References}}$
> - Bertsimas, Dimitris, Shimrit Shtern, and Bradley Sturt. "Two-stage sample robust optimization." Operations Research 70.1 (2022): 624-640

---

> ### Author Response · Authors · 2024-11-25
>
> Dear Reviewer q1qq
>
> Thank you for your thoughtful feedback.
>
> We would appreciate it if you could let us know whether our response addresses your concerns.
> If so, we kindly ask you to consider raising the rating of our work.
>
> Thank you again for your time and effort! We are looking forward to discussing any additional concerns you may have.
>
> Best wishes,
>
> Authors

---

### Meta-Review · Area_Chair_tT3Q · 2024-12-16

**Metareview:**

This paper introduces finite-sample guarantees for a robust optimization procedure for estimation in linear dynamical systems. The strengths are
(1) the method givens a clear and concise mathematical analysis of the method, demonstrating consistency
(2) The method demonstrates improvements over standard LSE on low data regimes
Weaknesses include
(1) there is no *theoretical justification* for the improvements of robust LSE over the standard LSE. Consequently, all benefits are purely empirical.
(2) given that the benefits are purely empirical, more experimentation would be preferrable.
(3) one of the reviewers noted that the authors do not extend their findings to misspecified models, which is one of the motivations for robustness.

Ultimately, this paper crosses the acceptance threshold, but its limitations preclude nomination from any special designation.

**Additional Comments On Reviewer Discussion:**

Unfortunately, and despite my prodding, authors were unresponsive during the paper discussion. My assessments above are based on my close reading of all reviews, and personal expertise in the field.

---

### Decision · Program_Chairs · 2025-01-22

Accept (Poster)